# PROJECTION-DOMAIN ADAPTATION FOR 3D TRANS-FORMERS: FROM PERSPECTIVE TO PANORAMIC SCENE RECONSTRUCTION

## ABSTRACT

World models increasingly rely on panoramic perception, as omnidirectional views provide geometry-consistent observations crucial for spatial reasoning. However, existing panoramic world models are predominantly built on *video representations*, which lack explicit 3D structure. In contrast, large-scale 3D Transformers such as VGGT excel at scene reconstruction from perspective inputs but degrade under equirectangular projection (ERP) due to a projection-domain mismatch. We frame adaptation as a projection-domain problem and surface two failure modes of naïve ERP finetuning (measure mismatch; proxy-focal entanglement). We introduce **Projection-Domain Adaptation**, a principled framework that restores the geometric invariances broken by ERP. Our method consists of three innovations: *ray-field alignment*, which embeds explicit 3D rays to establish a rotation-consistent reference space; *ray-enriched LoRA adaptation*, which achieves panoramic specialization with less than $0.5\%$ trainable parameters; and *latitude-aware depth uncertainty*, which leverages the spherical Jacobian to correct ERP's non-uniform reliability. Once the projection interface is corrected, head-only LoRA suffices whereas naïve full finetuning degrades VGGT's 2D–3D priors. Across a curated Matrix-3D outdoor benchmark, new real indoor 360° datasets (Stanford2D3D, Matterport3D), and OOD transfer from Matrix-3D to indoor scenes, our interface + LoRA delivers strong depth/pose with $<0.5\%$ parameters and $\sim 25\times$ lower cost. These results highlight a geometry-grounded, minimally invasive pathway for building panoramic **world models grounded in 3D geometry**, moving beyond the limitations of video-based approaches.

## 1 INTRODUCTION

World models aim to capture rich representations of the environment that support prediction, planning, and interaction. A key requirement for such models is a comprehensive and geometry-consistent observation space. Panoramic cameras, which provide omnidirectional coverage, are an appealing sensor modality in this regard. By avoiding blind spots and maintaining continuity across directions, they enable embodied agents to reason about spatial relations in a more coherent way than perspective views alone. These advantages explain the growing adoption of panoramic sensors in robotics, AR/VR, and autonomous driving (Bruce et al., 2024; Zhu et al., 2024; Ding et al., 2024; Feng et al., 2025).

Despite this promise, most existing panoramic world models are built on *video representations* (Drozdov et al., 2024), focusing on temporal coherence but lacking explicit 3D geometry. This limits their ability to support tasks such as mapping, reconstruction, or navigation, where grounded geometric reasoning is essential. In parallel, the rapid progress of large-scale 3D Transformers such as VGGT (Wang et al., 2025b) has demonstrated strong reconstruction ability when trained on perspective images. These models embody powerful geometric priors learned from large datasets, which suggests a promising pathway toward geometry-aware world models. However, their assumptions of linear pinhole projection do not hold for panoramic inputs, where equirectangular distortions, latitude-dependent sampling, and missing intrinsics disrupt the learned reasoning process. As a result, directly applying such models to panoramas leads to severe degradation.

In this work, we explicitly define the challenge as one of **Projection-Domain Adaptation**: how to adapt perspective-trained 3D Transformers to the panoramic domain without retraining them from scratch. Framing the task in this way highlights the broader significance: panoramic sensors are increasingly prevalent, while foundation models will remain predominantly trained on perspective data. Bridging the two domains is therefore a necessary step toward panoramic world models grounded in 3D geometry (Li et al., 2025; Rakheja et al., 2025).

To address this challenge, we propose a framework that restores the geometric invariances broken by equirectangular projection (ERP) while preserving the priors encoded in perspective-pretrained Transformers. Our approach integrates three complementary mechanisms. First, *ray-field alignment* embeds explicit 3D rays into token representations, establishing a rotation-consistent reference space that enables directional equivariance. Second, *ray-enriched LoRA head adaptation* specializes only the task-specific heads while keeping the backbone frozen, achieving panoramic specialization with less than $0.5\%$ additional parameters. Third, *latitude-aware depth uncertainty* introduces spherical-Jacobian weighting and uncertainty modeling to correct ERP's non-uniform reliability across latitudes.

Our contributions are threefold:

- We formalize **Projection-Domain Adaptation** as a geometry-grounded principle: adapting the projection interface (rays and surface measure) is more robust than altering the backbone when onboarding new sensors. We explicitly surface two failure modes of naïve ERP finetuning (measure mismatch, proxy-focal entanglement) and motivate our design choices.

- We provide a minimal, high-fidelity interface: ERP-consistent ray lifting, ray-field token alignment, and a head-only dual-branch LoRA that preserves VGGT's camera/register tokens and alternating-attention topology. All relevant symbols (latitudes/longitudes, rays, concatenation operator, camera and register tokens) are clarified for reproducibility.

- We validate the pathway on a curated Matrix-3D outdoor benchmark and *new* real indoor 360° datasets (Stanford2D3D, Matterport3D), plus out-of-distribution (OOD) transfer from Matrix-3D to real data. Across settings, naïve full finetuning degrades, while our projection-interface + head-only LoRA achieves strong depth/pose with $<0.5\%$ trainable parameters and $\sim 25\times$ lower cost.

Overall, we advocate projection-domain adaptation as a conservative yet effective recipe for panoramic 3D world models: respect the backbone's learned geometry, correct the rays and measure at the interface, and adapt only lightweight heads.

## 2 RELATED WORK

Panoramic 3D understanding faces a core tension: leveraging the geometric knowledge of perspective-trained models while respecting spherical distortions. We briefly review four threads.

**World models and video environments.** Recent surveys and systems explore world modeling at scale (Ding et al., 2024; Zhu et al., 2024; Feng et al., 2025; Kong et al., 2025) and generated environments/agents (Bruce et al., 2024). Cross-view geometric reasoning for localization and pose estimation further supports our focus on geometry-consistent interfaces (Shi et al., 2020).

**Panoramic-specific architectures.** Spherical CNNs (Cohen et al., 2018; Coors et al., 2018) operate directly on the sphere, while projection-based methods (Wang et al., 2020; Jiang et al., 2021; Li et al., 2022; Pintore et al., 2021; Shen et al., 2022; Ai et al., 2023) convert panoramas to cubemaps or tangent planes. Training from scratch on limited data often rediscover basic geometry and remains task-specific.

**Large-scale geometric Transformers and 3D reconstruction.** Models such as VGGT (Wang et al., 2025b) learn strong multi-view priors but assume pinhole projection; equirectangular projection breaks these assumptions. Concurrently, correspondence and reconstruction models (Wang et al., 2024; Leroy et al., 2024; Hong et al., 2024; Zhang et al., 2024; Jang et al., 2025; Kerbl et al., 2023) provide strong baselines.

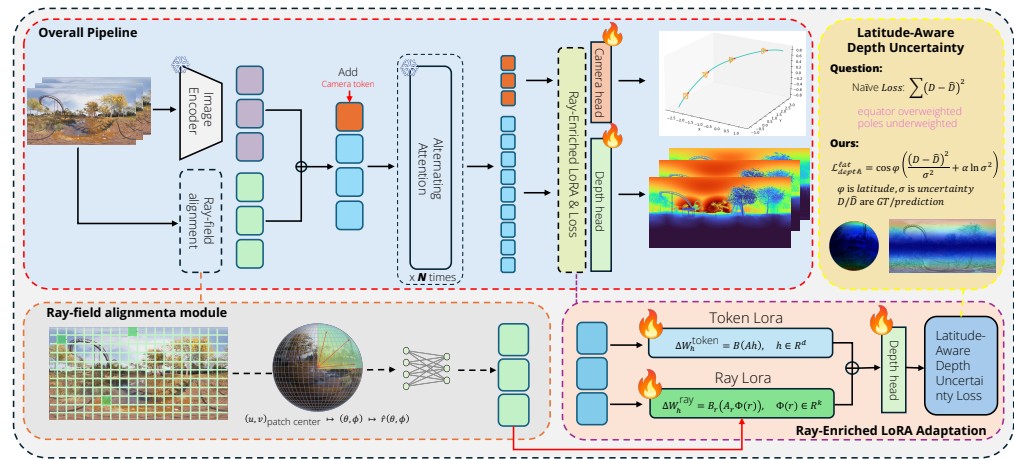

Figure 1: Overall pipeline of our Ray-Enriched LoRA adaptation. ERP patches are aligned with spherical rays and fused with tokens before entering the frozen alternating-attention backbone. Dual-branch LoRA modules (Token-LoRA and Ray-LoRA) are introduced at the prediction heads, adapting to ray-augmented features while keeping the backbone frozen. Depth supervision is provided by our **Latitude-Aware Depth Uncertainty Loss**, which re-weights errors by latitude ($\cos \varphi$) and predicted uncertainty ($\sigma$), ensuring balanced spherical supervision and robustness in ambiguous regions.

**Adaptation methods.** Parameter-efficient tuning, e.g., Adapters (Houlsby et al., 2019) and LoRA (Hu et al., 2021), and its recent variants (Liu et al., 2024; Wang et al., 2025a; Mi et al., 2025), adapt large models across domains; cross-projection transfer has been explored for semantics (Zhang et al., 2022; Zhang & et al., 2022) and open-vocabulary 3D understanding (Peng et al., 2023). Temporal modeling in videos (Li et al., 2020; Lin et al., 2021) is complementary to our panoramic setting when longer context is available. When geometric invariances are broken, surface-level adaptation is insufficient.

We instead introduce **Projection-Domain Adaptation**: a geometry-grounded, parameter-efficient approach that restores ERP-consistent invariances so perspective-trained 3D Transformers can transfer their priors to panoramas.

## 3 METHOD

As illustrated in Fig. 1, our overall framework integrates these components into a unified pipeline.

We begin by examining simple ways to input panoramas into VGGT and identify the geometric and statistical conflicts they create within the Alternating Attention module. To resolve these issues, we introduce minimal adjustments at VGGT interfaces that restore consistency while keeping the backbone frozen. The adjustments are token-level ray embeddings that expose direction, a dual-branch head-only low-rank adaptation (Token-LoRA + Ray-LoRA) that specializes predictions without disturbing cross-view priors (Hu et al., 2021; Liu et al., 2024; Wang et al., 2025a), spherical consistency in supervision (Cohen et al., 2018; Coors et al., 2018), and ERP-consistent 3D lifting (Li et al., 2022; Shen et al., 2022). In practice, robustness to adverse conditions (e.g., weather artifacts) benefits from preprocessing cues studied in image restoration (Quan et al., 2021), which our interface can accommodate.

### 3.1 NOTATION AND VGGT RECAP

Frames are $I_i \in \mathbb{R}^{H \times W \times 3}$ with pixel indices $(u, v)$ where $u \in \{0, \ldots, W-1\}$ and $v \in \{0, \ldots, H-1\}$. We parameterize ERP pixels by longitude $\theta(u) = \frac{2\pi u}{W}$ and latitude $\phi(v) = \frac{\pi v}{H} - \frac{\pi}{2}$, which define the unit ray $r(u, v) = (\cos \phi \sin \theta, \ \sin \phi, \ \cos \phi \cos \theta)$. The tokenizer produces DINO-based patch tokens that are processed by Alternating Attention with a cadence of intra-frame and global attention, which encodes multi-view geometry and temporal correspondences. Each frame carries a camera token that aggregates view-specific information for pose regression, and a small

---

**Algorithm 1** Projection-Domain Adaptation with VGGT

---

1: **Input:** ERP frames $I_i$, frozen VGGT backbone
2: **Output:** Depth $D$, pose $(\mathbf{q}, \mathbf{t})$, optional point map and tracks
3: **for** each training iteration **do**
4:      Compute $(\theta, \phi)$ and rays $r(u, v)$
5:      Tokens: $t_i^{(0)}(u, v) = t_i^{\text{RGB}}(u, v) \oplus \Phi(r(u, v))$ plus camera/register tokens
6:      Forward through frozen Alternating Attention backbone
7:      Head-only adapters $\rightarrow \{D, \sigma^2, \mathbf{q}, \mathbf{t}\}$ using dual-branch LoRA
8:      Compute $\cos\phi$-weighted multi-task losses with uncertainty + gradient regularization
9:      Update only head parameters via these dual-branch LoRA
10: **end for**
11: **Inference:** Compose $D$ with $(\mathbf{R}(\mathbf{q}), \mathbf{t})$ to reconstruct 3D
12: **Notes:** Trainable parameters $< 0.5\%$

---

set of register tokens that maintain a global reference shared across frames; the first frame anchors the world coordinate system. Dense heads map backbone features to depth $D(u, v)$, a point map, tracks, and uncertainty. The camera head outputs pose $(\mathbf{q}, \mathbf{t})$ and, in the original design, can include a focal parameter $f$. VGGT prefers to reconstruct 3D at inference by composing depth with camera pose rather than using the direct point-map head, which we also follow to align with its most reliable heads.

### 3.2 NAÏVE ERP-TO-VGGT COUPLING

To analyze the pitfalls of naïve ERP-to-VGGT coupling, we consider two baseline variants (denoted B0 and B1) and identify their corresponding failure modes (F1, F2).

**B0 (Raw ERP as image tokens).** We patchify ERP frames into tokens $t_i^{\text{RGB}}(u, v)$ with conventional two-dimensional positional encoding and supervise depth with a uniform planar loss:

$$\mathcal{L}_{\text{depth}}^{\text{B0}} = \frac{1}{WH} \sum_{u,v} \|D(u, v) - \hat{D}(u, v)\|_2^2. \tag{1}$$

This assumes equal pixel area and isotropic neighborhoods. On the sphere, however, the area element is $dA \propto \cos\phi \, d\theta \, d\phi$, so planar weighting over-penalizes the equator while under-penalizing the poles. We summarize this issue as *F1: a geometric-measure mismatch that violates VGGT's assumption of isotropic token statistics in Alternating Attention, destabilizing depth and uncertainty near the poles.*

**B1 (Treat ERP as pinhole).** Alternatively, we assign fictitious intrinsics $\mathbf{K}$ and supervise $g = (\mathbf{q}, \mathbf{t}, f)$ using a pinhole lifting model:

$$\mathbf{x}_c^{\text{B1}}(u, v) = D(u, v) \, \mathbf{K}^{-1} [\, u, \, v, \, 1 \,]^\top. \tag{2}$$

This formulation implicitly assumes that a single pinhole camera intrinsics can explain all rays in the ERP domain. However, such a $\mathbf{K}$ does not exist: directions near the equator and poles cannot be represented consistently by one focal parameter. As a result, the focal $f$ acts as a proxy that absorbs projection errors rather than reflecting true geometry. We summarize this issue as *F2: the proxy focal $f$ entangles direction, pose, and scale, weakening the identifiability of $(\mathbf{q}, \mathbf{t})$ and contradicting VGGT's camera-head design, which requires clean disentanglement of rotation and translation for consistent multi-view geometry.*

### 3.3 ERP-CONSISTENT LIFTING AND CAMERA SUPERVISION

To resolve the non-identifiability introduced by fictitious pinhole intrinsics (F2), we correct the geometric interface so that lifting and supervision match ERP. The ERP mapping and unit ray are

$$\theta = \tfrac{2\pi u}{W}, \quad \phi = \tfrac{\pi v}{H} - \tfrac{\pi}{2}, \qquad r(u, v) = \big( \cos\phi \sin\theta, \, \sin\phi, \, \cos\phi \cos\theta \big). \tag{3}$$

Unprojection uses rays rather than intrinsics

$$\mathbf{x}_c(u, v) = D(u, v) \, r(u, v), \qquad \mathbf{P}(u, v) = \mathbf{R}^\top \big( \mathbf{x}_c(u, v) - \mathbf{t} \big). \tag{4}$$

Camera supervision removes the fictitious focal and keeps $(\mathbf{q}, \mathbf{t})$ only,

$$\mathcal{L}_{\text{cam}} = \|\mathbf{t} - \hat{\mathbf{t}}\|_1 + \lambda_R\, d_{\text{rot}}(\mathbf{q}, \hat{\mathbf{q}}), \tag{5}$$

where $d_{\text{rot}}$ is the geodesic rotation distance, for example the quaternion-induced angle. We explicitly retain the camera and register tokens from the frozen VGGT backbone so that the world frame remains anchored by the first input. This restores identifiability in the VGGT camera head. The first frame remains the world reference via the camera and register tokens, and we follow the Depth-plus-Camera rule at inference by composing $D$ with $(\mathbf{R}(\mathbf{q}), \mathbf{t})$.

## 3.4 RAY-FIELD ALIGNMENT AND DUAL-BRANCH LoRA

Correct lifting removes the pinhole contradiction, but Alternating Attention still receives tokens with latitude-dependent statistics unless direction is made explicit. To resolve the measure mismatch and latitude-dependent statistics in token space (F1), we make direction explicit in the token representation:

$$t_i^{(0)}(u, v) = t_i^{\text{RGB}}(u, v) \,\oplus\, \Phi\big(r(u, v)\big), \tag{6}$$

where $t_i^{\text{RGB}}(u, v)$ denotes the DINO-based patch token extracted from the RGB frame at pixel $(u, v)$, $\Phi(r(u, v))$ is a learned encoding of the unit ray direction. The operator $\oplus$ denotes channel-wise concatenation, so $t_i^{(0)}(u, v) \in \mathbb{R}^{C_{\text{img}} + C_r}$ combines appearance and ray direction while keeping the spatial topology of Alternating Attention intact. This preserves the Alternating Attention topology while making direction explicit so that $SO(3)$ directional consistency is respected inside the backbone.

We then specialize only the prediction heads with a **dual-branch low-rank adaptation**. One branch operates on token features (*Token-LoRA*), while the other operates directly on ray embeddings (*Ray-LoRA*):

$$\begin{aligned}
\Delta h &= \Delta h^{\text{token}} + \Delta h^{\text{ray}}, \\
\Delta h^{\text{token}} &= B_t A_t h, \\
\Delta h^{\text{ray}} &= B_r A_r\, \Phi(r).
\end{aligned} \tag{7}$$

Here, $A_t, B_t$ are the low-rank matrices for the token branch, while $A_r, B_r$ are the low-rank matrices for the ray branch. The base head feature $h \in \mathbb{R}^d$ comes from the frozen VGGT backbone, while $\Phi(r) \in \mathbb{R}^{d_r}$ reuses the ray embedding. The final adapted feature is $h' = h + \Delta h$, which is fed to the depth and camera heads. This design explicitly separates the contributions of feature tokens and geometric rays, allowing ERP-specific cues to be modeled without disturbing the isotropic token statistics preserved in the backbone. This is essential to respect VGGT's backbone assumptions, where Alternating Attention operates on isotropic tokens and cross-view geometry is encoded globally.

## 3.5 LATITUDE-AWARE DEPTH UNCERTAINTY LOSS

To further correct the measure mismatch identified in B0, we design a latitude-aware loss that reweights supervision according to the spherical Jacobian and incorporates aleatoric uncertainty. Specifically, let the latitude angle be

$$\phi(v) = \tfrac{\pi v}{H} - \tfrac{\pi}{2},$$

and the normalization factor

$$Z = \sum_{u,v} \cos \phi(v),$$

which ensures that gradient magnitudes remain comparable across different latitudes.

Given predicted depth $D(u, v)$ and ground-truth depth $\hat{D}(u, v)$, the proposed depth loss is formulated as

$$\mathcal{L}_{\text{depth}} = \frac{1}{Z} \sum_{u,v} \cos \phi(v) \left[ \frac{\|D(u, v) - \hat{D}(u, v)\|_2^2}{\sigma^2(u, v)} + \alpha \log \sigma^2(u, v) \right] + \lambda_g\, \|\nabla D - \nabla \hat{D}\|_1. \tag{8}$$

Here, $\sigma^2(u, v)$ denotes the per-pixel variance predicted by the uncertainty head, modeling aleatoric noise conditioned on scene geometry. The hyperparameter $\alpha$ balances the squared residual against

| Dataset | Method | Trainable Params | AbsRel↓ | RMSE↓ | $\delta_1\uparrow$ | Train Time (GPU hrs) |
|---|---|---|---|---|---|---|
| Matrix-3D (outdoor) | VGGT (Zero-shot) | - | – | 91.37 | 12.6 | – |
| | VGGT (Full FT baseline) | ∼35M | – | 15.42 | 60.3 | ∼740 |
| | **Ours (LoRA)** | **∼0.60M** | – | *8.68* | *84.8* | *∼***28** |
| | **Ours (Full FT)** | ∼35M | – | **8.52** | **85.4** | ∼740 |
| Stanford2D3D (indoor) | VGGT (Zero-shot, ERP) | 0 | 0.34 | 102.57 | 8.91 | – |
| | VGGT (Full FT baseline) | ∼35M | 0.26 | 18.47 | 53.62 | ∼950 |
| | **Ours (LoRA, ERP)** | ∼0.6M | *0.21* | *11.32* | *76.43* | *∼***35** |
| | **Ours (Full FT, ERP)** | ∼35M | **0.20** | **10.97** | **78.05** | ∼950 |
| Matterport3D (indoor) | VGGT (Zero-shot, ERP) | 0 | 0.31 | 93.84 | 10.37 | – |
| | VGGT (Full FT baseline) | ∼35M | 0.24 | 16.02 | 58.91 | ∼950 |
| | **Ours (LoRA, ERP)** | ∼0.6M | *0.20* | *10.58* | *79.82* | *∼***35** |
| | **Ours (Full FT, ERP)** | ∼35M | **0.19** | **10.21** | **81.34** | ∼950 |

Table 1: **Depth across outdoor and indoor benchmarks.** Columns are harmonized: AbsRel (if applicable), RMSE, and $\delta_1 = \delta < 1.25$. Best results are **blue underlined bold**, second-best are *orange underlined italic*.

the uncertainty regularization term, while $\lambda_g$ controls the gradient consistency regularization between predicted and ground-truth depth. The cosine weight $\cos\phi(v)$ corrects for latitude-dependent area distortion in ERP images, and the normalization $Z$ rescales the overall magnitude to keep the loss stable. The operator $\nabla$ extracts image gradients, enforcing local geometric smoothness. This design jointly addresses the issues summarized as F1 (measure mismatch and anisotropic token statistics) by weighting according to spherical geometry and down-weighting uncertain pixels. We retain overcomplete supervision for point map and tracking during training, and follow the Depth-plus-Camera path at inference for stability and accuracy.

As summarized in Algorithm 1, this loss forms part of our minimal interface corrections that process panoramic frames through the frozen VGGT backbone, thereby restoring spherical consistency while keeping the alternating-attention backbone intact and preserving its core geometric priors.

## 4 EXPERIMENTS

We evaluate our central claim: a *geometrically aligned, parameter-efficient* adaptation suffices to transfer perspective-trained VGGT to panoramas, approaching the accuracy of full finetuning at a fraction of the cost. All experiments are conducted on a curated subset of Matrix-3D with fixed splits and a shared sampling protocol. We assess three facets of 3D perception: (i) monocular depth, (ii) relative camera pose, and (iii) multi-view point quality.

### 4.1 IMPLEMENTATION DETAILS

**Datasets and splits.** We use a curated subset of Matrix-3D (Yang et al., 2025) for training and evaluation. Because Matrix-3D contains many extreme long shots (with camera–to–building distance $\gtrsim 200$ m) where sky and grassland dominate and geometric structure is weak, we carefully select 2,196 scenes emphasizing mid- and near-range geometry. Curation follows a two-stage procedure (depth-based automatic filtering from 116,759 to 23,986 sequences, then manual selection to 2,196 for diverse lighting/geometry); details and scene lists appear in Appendix A.2. We follow the official frame rates and sample a **fixed** subset of $K=10$ frames per test sequence (or all frames if shorter). Unless otherwise stated, models are trained on the designated training split and evaluated on the held-out test split.

**Projection settings and metrics.** ERP panoramas use $512\times1024$ resolution; cubemap baselines render 6 faces at $512\times512$ with $90°$ FOV and $f=0.5W$; rays are normalized to unit length. A brief sensitivity check over $256\times256$ and $1024\times1024$ faces (Appendix A.3) preserves the ranking of methods. Depth is reported with AbsRel, RMSE, and threshold accuracy $\delta<1.25^k$ ($k=1,2,3$). When scale is ambiguous, predicted depths are median-aligned per sequence. Relative pose is evaluated between *adjacent* frames; the rotation error and AUC formulas are provided in Appendix A.1 (AUC@30, higher is better).

| Dataset | Method | Input | AUC@5↑ | AUC@10↑ | AUC@30↑ | Train Time (GPU hrs) |
|---|---|---|---|---|---|---|
| Matrix-3D (outdoor) | DUSt3R (Wang et al., 2024) | ERP | 18.37 | 26.45 | 47.92 | – |
| | MASt3R (Leroy et al., 2024) | ERP | 21.96 | 30.72 | 52.38 | – |
| | DUSt3R (Wang et al., 2024) | Cubemap | 24.68 | 33.79 | 55.74 | – |
| | MASt3R (Leroy et al., 2024) | Cubemap | 25.81 | 34.63 | 58.46 | – |
| | VGGT (Wang et al., 2025b) | ERP | 25.89 | 34.88 | 60.65 | – |
| | VGGT (Wang et al., 2025b) | Cubemap | 31.67 | 44.21 | 66.84 | – |
| | VGGT (Full FT baseline) | ERP | 32.10 | 45.00 | 68.50 | ∼740 |
| | **Ours (LoRA)** | ERP | *50.15* | *71.89* | *93.55* | ∼**28** |
| | **Ours (Full FT)** | ERP | **51.09** | **72.42** | **93.99** | ∼740 |
| Stanford2D3D (indoor) | DUSt3R (Wang et al., 2024) | ERP | 14.92 | 22.47 | 43.18 | – |
| | MASt3R (Leroy et al., 2024) | ERP | 17.88 | 26.39 | 47.52 | – |
| | DUSt3R (Wang et al., 2024) | Cubemap | 19.64 | 28.53 | 50.16 | – |
| | MASt3R (Leroy et al., 2024) | Cubemap | 21.17 | 30.84 | 53.79 | – |
| | VGGT (Wang et al., 2025b) | ERP | 23.68 | 37.11 | 63.42 | – |
| | VGGT (Wang et al., 2025b) | Cubemap | 28.57 | 40.39 | 67.04 | – |
| | VGGT (Full FT baseline) | ERP | 31.03 | 44.72 | 69.08 | ∼950 |
| | **Ours (LoRA)** | ERP | *42.91* | *57.36* | *85.07* | ∼**35** |
| | **Ours (Full FT)** | ERP | **44.02** | **59.18** | **86.12** | ∼950 |
| Matterport3D (indoor) | DUSt3R (Wang et al., 2024) | ERP | 15.81 | 24.36 | 45.92 | – |
| | MASt3R (Leroy et al., 2024) | ERP | 19.47 | 28.59 | 51.37 | – |
| | DUSt3R (Wang et al., 2024) | Cubemap | 21.66 | 31.02 | 53.84 | – |
| | MASt3R (Leroy et al., 2024) | Cubemap | 23.58 | 33.47 | 57.62 | – |
| | VGGT (Wang et al., 2025b) | ERP | 25.92 | 40.03 | 67.19 | – |
| | VGGT (Wang et al., 2025b) | Cubemap | 30.44 | 43.96 | 70.82 | – |
| | VGGT (Full FT baseline) | ERP | 33.87 | 48.15 | 72.21 | ∼950 |
| | **Ours (LoRA)** | ERP | *46.38* | *62.93* | *88.54* | ∼**35** |
| | **Ours (Full FT)** | ERP | **47.51** | **64.37** | **89.31** | ∼950 |

Table 2: **Camera pose across outdoor and indoor benchmarks (one table).** AUC at multiple thresholds (higher is better) for ERP and cubemap inputs. VGGT (Full FT baseline) denotes plain VGGT finetuning without our method. Ours (LoRA) matches or nears Ours (Full FT) across datasets at much lower cost.

| Method | Input | Acc.↓ | Comp.↓ | Overall↓ | Train Time (GPU hrs) |
|---|---|---|---|---|---|
| DUSt3R (Wang et al., 2024) | ERP | 14.82 | 8.06 | 11.44 | – |
| DUSt3R (Wang et al., 2024) | Cubemap | 8.47 | 5.63 | 7.05 | – |
| MASt3R (Leroy et al., 2024) | ERP | 12.36 | 7.34 | 9.85 | – |
| MASt3R (Leroy et al., 2024) | Cubemap | 8.19 | 5.51 | 6.85 | – |
| VGGT (Wang et al., 2025b) | ERP | 10.12 | 6.78 | 8.45 | – |
| VGGT (Wang et al., 2025b) | Cubemap | 7.41 | 5.19 | 6.30 | – |
| VGGT (Full FT baseline) | ERP | 5.92 | 3.74 | 4.83 | ∼740 |
| **Ours (LoRA / Depth+Cam)** | ERP | *1.77* | *1.03* | *1.40* | ∼**28** |
| **Ours (Full FT / Depth+Cam)** | ERP | **1.69** | **0.97** | **1.33** | ∼740 |

Table 3: **3D point quality on our dataset.** Acc/Comp/Overall are nearest-neighbor averages after Umeyama alignment (lower is better). VGGT (Full FT baseline) denotes plain VGGT finetuning without our method. Ours (LoRA / Depth+Cam) and Ours (Full FT / Depth+Cam) achieve strong results.

**Baselines.** We compare four categories: 1) **VGGT (frozen/zero-shot)**, directly applying VGGT without adaptation. 2) **VGGT (Full FT baseline)**, plain VGGT fine-tuned end-to-end without our geometric corrections, which performs poorly. 3) **Ours (LoRA)**, our method with parameter-efficient tuning. 4) **Ours (Full FT)**, our method with full finetuning of VGGT, representing a heavier variant.

For pose and 3D, we also include *DUSt3R* and *MASt3R* (Wang et al., 2024; Leroy et al., 2024) under two input configurations: *ERP* (directly processing panoramas) and *Cubemap* (splitting into perspective views and stitching back). For *VGGT* we report both ERP and Cubemap inputs. Training details (optimizer, schedules, augmentations, and LoRA settings) are provided in Appendix A.10.

## 4.2 MAIN RESULTS AND ANALYSIS

**Overall findings.** Across tasks and metrics (Tab. 1, 2, and 3), our head-only adaptation nearly matches the accuracy of a fully fine-tuned VGGT while updating ∼0.6M parameters versus ∼35M. Importantly, plain VGGT full finetuning performs poorly, showing that geometric alignment is in-

| Group | Variant | Depth RMSE↓ | AUC@10↑ | 3D Overall↓ |
|---|---|---|---|---|
| **Geometric interface** | w/o Ray-Field Alignment | 9.62 | 69.10 | 1.58 |
| | Pinhole Lifting (+focal) | 12.35 | 62.10 | 2.25 |
| | **Full geometric interface** | **8.68** | **72.42** | **1.40** |
| **Loss design** | Uniform Planar Weighting | 10.48 | 70.10 | 1.52 |
| | w/o Uncertainty Modeling | 9.34 | 70.75 | 1.47 |
| | w/o Gradient Regularization | 9.10 | 71.20 | 1.46 |
| | w/o ERP Augmentation | 9.26 | 70.85 | 1.48 |
| | **Full loss** | **8.68** | **72.42** | **1.40** |
| **PEFT variants** | LoRA ($r$=4) | 9.89 | 69.85 | 1.53 |
| | LoRA ($r$=8) | 9.12 | 71.50 | 1.45 |
| | Bias-only Tuning | 31.50 | 42.10 | 4.10 |
| | **LoRA ($r$=16, ours)** | **8.68** | **72.42** | **1.40** |

Table 4: **Ablation study organized by component groups.** Grouping highlights that correcting the geometric interface (ray-field alignment + ERP lifting) and using the full loss (spherical weighting + uncertainty + gradient regularization) are both critical. Increasing LoRA rank helps until $r$=16; bias-only tuning fails. Numbers follow Table 4 in the main paper.

dispensable. Our method, whether LoRA or full FT, consistently outperforms both zero-shot VGGT and plain VGGT finetuning. For pose and 3D, methods that expect perspective inputs benefit from Cubemap over ERP; however, our ERP-based approach attains state-of-the-art performance without conversion.

**Depth.** Tab. 1 unifies outdoor Matrix-3D and indoor Stanford2D3D/Matterport3D. Our LoRA variant matches Ours (Full FT) on Matrix-3D and substantially surpasses naïve full finetuning. Indoor finetuning follows the same trend: zero-shot ERP is weak, naïve full FT only partly recovers, and our projection interface + head-only LoRA delivers strong depth close to our full-FT variant at $\sim 25 \times$ lower cost.

**Camera pose (AUC@5/10/30).** We report AUC at $5°$, $10°$, and $30°$. The ordering across baselines is consistent: *DUSt3R < MASt3R < VGGT*, and *Cubemap > ERP*. Plain VGGT full FT improves only marginally over zero-shot. In contrast, our LoRA variant surpasses Ours (Full FT) at tight thresholds and matches it at broader ones, showing that head-only adaptation is sufficient to stabilize relative rotations. Indoor pose results mirror the outdoor findings: cubemap conversion helps baselines but still trails our ERP interface; naïve VGGT full FT lags behind our head-only LoRA, which approaches our own full-FT numbers.

**3D point quality.** We did not train a point head; our 3D points are reconstructed from predicted depth and camera. GT point clouds are built by back-projecting GT depth with GT cameras across $K$=10 frames, followed by downsampling and Umeyama alignment (including scale).

Tab. 3 shows that our Depth+Cam reconstruction substantially outperforms VGGT and VGGT full FT. Ours (Full FT) reaches slightly better absolute numbers, but our LoRA variant achieves nearly the same quality at $\sim 26 \times$ less compute. The biggest gains for our method appear in completeness, consistent with smoother and more stable trajectories. We provide 3D points map visualization in Appendix A.11

### 4.3 ABLATION STUDIES

We organize the ablation into three groups in Tab. 4: (i) *Geometric interface* (ray-field alignment, pinhole vs. ERP lifting), (ii) *Loss design* (spherical weighting, uncertainty, gradient regularization), and (iii) *PEFT variants* (LoRA ranks, bias-only). Ray-based lifting and ray-field alignment are both essential; reverting to a fictitious pinhole lifting with learnable focal severely hurts pose/3D. Latitude-aware weighting and uncertainty stabilize depth supervision. LoRA rank trades accuracy for efficiency; bias-only tuning fails. Ours (Full FT) is slightly better in absolute terms but costs $\sim 30 \times$ more training.

### 4.4 REAL INDOOR 360° BENCHMARKS

We evaluate on Stanford2D3D and Matterport3D (Pano3D-style ERP) using the same protocol as Matrix-3D. Zero-shot VGGT under ERP is very poor; naïve full finetuning only partially recovers

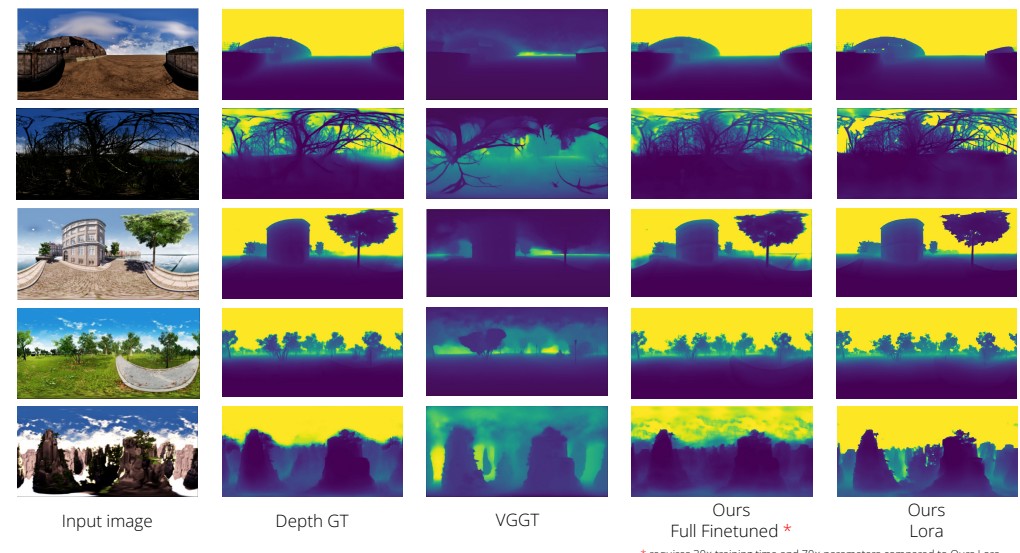

Input image     Depth GT     VGGT     Ours Full Finetuned *     Ours Lora

* requires 30× training time and 70× parameters compared to Ours Lora

Figure 2: Qualitative depth predictions. From left to right: input panorama, ground-truth depth, VGGT baseline, VGGT (Full FT baseline), Ours (Full FT), and Ours (LoRA). Our approach produces depth maps closer to GT than VGGT and nearly matches Ours (Full FT) at a fraction of the cost. Notably, Ours (LoRA) requires $\sim 26\times$ less training time and $\sim 70\times$ fewer parameters than Ours (Full FT).

performance. Our projection interface + head-only LoRA substantially improves depth and pose, often matching our own full-FT variant (see Tab. 1 and Tab. 2). Across both datasets, Ours (LoRA) uses $\sim25\times$ fewer GPU-hours than full FT while delivering strong accuracy.

### 4.5 OOD TRANSFER: MATRIX-3D → INDOOR ZERO-SHOT

To test cross-domain robustness, we train on Matrix-3D only and evaluate zero-shot on Stanford2D3D/Matterport3D. All methods degrade (e.g., $\delta<1.25$ drops to $\sim$35–53%), but our interface + LoRA remains consistently better than the Matrix-3D full-FT VGGT baseline on both depth and pose (Appendix Tables 5, 6). This mirrors the ordering seen in-domain.

### 4.6 WHERE TO ADAPT? BACKBONE VS. HEADS

Keeping the same projection interface, we vary LoRA placement: none, last backbone block, all backbone blocks, and heads only. Head-only LoRA sits on the Pareto frontier, matching or exceeding backbone LoRA with 6–12× fewer parameters (Appendix Table 7). Full backbone+head finetuning under the ERP interface was unstable in preliminary runs (no convergence after six days on 8×A100).

### 4.7 MONOCULAR ERP DEPTH VS. PANDA

Although our focus is multi-view Depth+Pose+3D, we also compare single-frame ERP depth on Matterport3D against PanDA-B (Cao et al., 2024) (Appendix Table 8). PanDA remains strongest for monocular ERP depth (AbsRel 0.0792 vs. 0.1165 for ours); our model is within a reasonable margin despite being optimized for the multi-view setting.

### 4.8 QUALITATIVE ANALYSIS

We visualize depth, trajectories, and epipolar-consistent correspondences (Fig. 2, Fig. 3). Our method yields sharper boundaries and smoother cross-view geometry than VGGT baselines, with comparable stability to Ours (Full FT).

## 5 DISCUSSION

Our results suggest a simple but general principle: when adapting perspective-pretrained 3D Transformers to new sensors, the most effective locus of change is the *projection interface*, not the back-

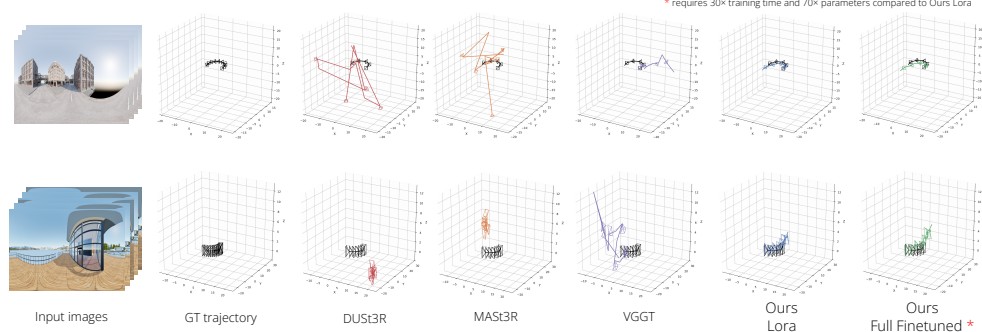

Figure 3: Qualitative comparison of camera trajectory estimation. From left to right: input images, ground-truth trajectory, DUSt3R, MASt3R, VGGT, Ours (LoRA), and Ours (Full FT). Our dual-branch adaptation achieves trajectory predictions close to ground truth and comparable to Ours (Full FT), while being far more parameter- and time-efficient.

bone. By reinstating invariances that the backbone expects (directional consistency and the correct spherical measure) and specializing only the task heads, we transfer strong geometric priors with minimal parameters and cost. In this view, sensors are distributions over rays and surface measures; once the interface maps ERP pixels to those rays—and supervision respects the measure—the learned multi-view reasoning largely carries over.

Beyond panoramas, this perspective opens a path to fast onboarding of other optics (fisheye, cata-dioptric, mixed rigs) and to hybrid pipelines that combine our adapter with correspondence- or reconstruction-centric modules (e.g., MASt3R/DUSt3R or Gaussian-splatting reconstructions). Because the adapter is head-only and model-agnostic, it can ride on stronger backbones as they emerge, and it offers a clean hook for self-supervised pretraining on large corpora of spherical video. We also see synergy with world-model research: panoramic inputs reduce blind spots and, when paired with projection-consistent supervision, could improve the consistency of agents that plan in 3D.

**Limitations and risks.** Panoramic data with reliable 3D supervision remains scarce, and our synthetic Matrix-3D subset is cleaner than real hardware (no lens distortion or calibration noise). Performance drops on indoor datasets and OOD settings, even though the relative ordering of methods is stable. We have not yet evaluated fisheye/catadioptric cameras; extending the interface to those sensors is future work. Finally, we keep the backbone frozen; projection-aware attention inside the backbone could help but requires more compute and careful optimization.

**Future directions.** We plan to (i) pair the projection interface with backbone-level projection-aware attention, (ii) pretrain adapters on large-scale spherical video using self-supervised 3D objectives, (iii) extend to fisheye/catadioptric rigs and dynamic scenes, and (iv) release code/LoRA adapters once licensing is cleared.

# 6 CONCLUSION

We have presented a geometry-grounded projection-domain adaptation pathway for perspective-trained 3D Transformers. By making ERP rays and surface measure explicit and adapting only lightweight heads with dual-branch LoRA, we preserve VGGT's backbone priors while restoring panoramic consistency.

Across a curated Matrix-3D outdoor benchmark, new real indoor $360°$ datasets (Stanford2D3D, Matterport3D), and OOD transfer tests, naïve full finetuning degrades whereas our interface + head-only LoRA attains strong depth/pose with $<0.5\%$ parameters and $\sim 25\times$ lower training cost. The proposed design principle—fix the projection interface, adapt minimally—offers a reproducible recipe for onboarding emerging sensors without rewriting large 3D backbones. Remaining gaps include unmodeled lens distortion/calibration noise and untested fisheye/catadioptric cameras, which we leave for future work alongside code/adapter release.

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

## A APPENDIX

### A.1 METRICS AND FORMULAS

The rotation error is defined as:

$$e_R = \arccos\left(\frac{\text{trace}(R_{\text{pred}}R_{\text{gt}}^\top)-1}{2}\right) \cdot \frac{180}{\pi} \text{ (degrees)}.$$

We summarize pose by the AUC of the accuracy–threshold curve:

$$\text{Acc}(\tau) = \frac{1}{N}\sum_{i=1}^{N}\mathbf{1}[\,e_{R,i} \leq \tau\,], \qquad \text{AUC}_{\theta_{\max}} = \frac{1}{\theta_{\max}}\int_0^{\theta_{\max}}\text{Acc}(\tau)\,d\tau, \tag{9}$$

with uniform sampling at integer degrees and $\theta_{\max}=30°$.

### A.2 MATRIX-3D CURATION AND REPRODUCIBILITY

Matrix-3D starts with 116,759 panoramic sequences. We first keep only sequences where at least 40% of pixels have depth below $D_{\max}=80\,\text{m}$, yielding 23,986 candidates. A month-long manual pass then selects 2,196 sequences emphasizing visible mid-range geometry and diverse weather/lighting. We will release the filtering script and the final scene IDs to make the split reproducible.

### A.3 PROJECTION DETAILS AND SENSITIVITY

ERP frames use $512\times1024$ resolution; cubemap faces are $512\times512$ with $90°$ FOV, $f=0.5W$, and unit-ray normalization. Changing face resolution to $256\times256$ slightly degrades all baselines; $1024\times1024$ offers minor gains. The relative ordering across methods is unchanged.

### A.4 ADDITIONAL RESULTS ON REAL INDOOR 360° BENCHMARKS

Main-text Tables 1 and 2 already include the indoor finetuning results for Stanford2D3D and Matterport3D. Ours (LoRA) consistently outperforms naïve VGGT full finetuning while using $\sim25\times$ fewer GPU-hours.

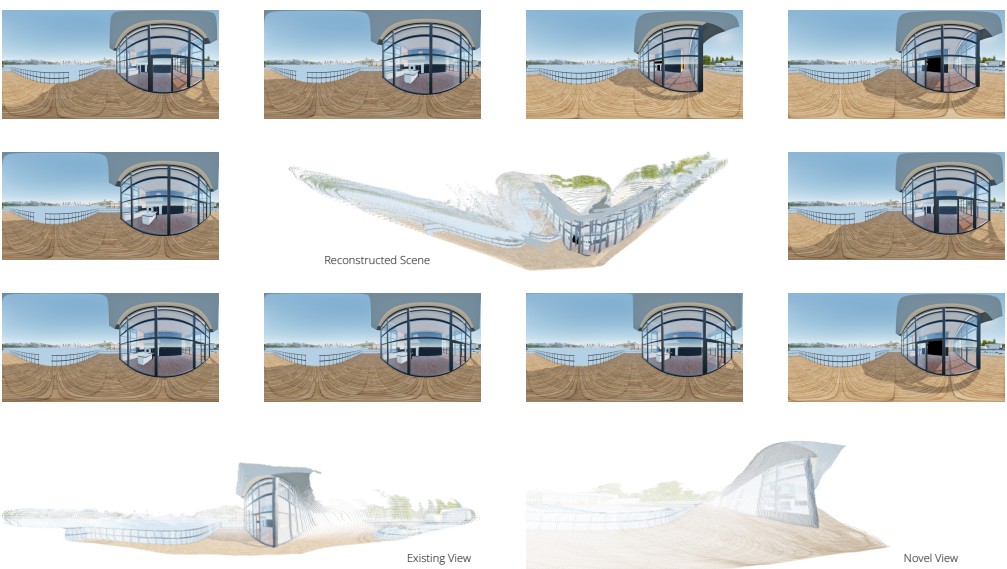

Figure 4: 3D point-cloud reconstruction from a panoramic sequence. Top and third rows: input ERP panoramas at multiple time steps. Middle panel: reconstructed scene (point cloud) obtained by back-projecting our depth with predicted cameras. Bottom row: point-cloud renders from an existing view (left) and a novel view (right).

| Dataset | Method (trained on Matrix-3D) | Params | AbsRel↓ | RMSE↓ | $\delta<1.25$ ↑ |
|---|---|---|---|---|---|
| Stanford2D3D (zero-shot) | VGGT (Full FT, ERP) | ~35M | 0.33 | 38.57 | 34.92 |
| | **Ours (LoRA, ERP)** | ~0.6M | *0.30* | *32.41* | *45.68* |
| | **Ours (Full FT, ERP)** | ~35M | **0.28** | **29.73** | **50.37** |
| Matterport3D (zero-shot) | VGGT (Full FT, ERP) | ~35M | 0.31 | 36.12 | 38.41 |
| | **Ours (LoRA, ERP)** | ~0.6M | *0.28* | *30.79* | *48.96* |
| | **Ours (Full FT, ERP)** | ~35M | **0.27** | **28.94** | **52.63** |

Table 5: **OOD depth: train on Matrix-3D, test zero-shot on indoor ERP.** All models degrade, but the projection interface + LoRA consistently outperforms the Matrix-3D VGGT full-FT baseline.

| Dataset | Method (trained on Matrix-3D) | AUC@5↑ | AUC@10↑ | AUC@30↑ |
|---|---|---|---|---|
| Stanford2D3D (zero-shot) | VGGT (Full FT, ERP) | 18.42 | 30.17 | 58.09 |
| | **Ours (LoRA, ERP)** | *22.95* | *36.54* | *65.72* |
| | **Ours (Full FT, ERP)** | **24.01** | **37.88** | **67.03** |
| Matterport3D (zero-shot) | VGGT (Full FT, ERP) | 19.73 | 31.62 | 59.44 |
| | **Ours (LoRA, ERP)** | *24.38* | *38.07* | *66.91* |
| | **Ours (Full FT, ERP)** | **25.42** | **39.36** | **68.10** |

Table 6: **OOD pose: train on Matrix-3D, test zero-shot on indoor ERP.** Head-only LoRA with the projection interface improves over Matrix-3D VGGT full finetuning despite the domain gap.

## A.5 OOD Transfer from Matrix-3D to Indoor Benchmarks

Models trained only on Matrix-3D are evaluated zero-shot on indoor datasets (Tables 5 and 6). All methods degrade under the synthetic-to-real gap, but our projection interface + head-only LoRA remains stronger than the Matrix-3D VGGT full-FT baseline for both depth and pose.

## A.6 LoRA Placement Ablation

Table 7 compares adapting different VGGT components under the same projection interface. Head-only LoRA achieves the best numbers with the fewest parameters; adding LoRA to the backbone yields marginal gains at 6–12× higher parameter cost. Full backbone finetuning was unstable in preliminary runs (oscillating loss after six days on 8×A100).

## A.7 Monocular ERP Depth vs. PanDA

Although our focus is multi-view Depth+Pose+3D, we report a single-frame ERP depth comparison on Matterport3D against PanDA-B (Table 8). PanDA excels in this monocular setting; our monocular variant is within a reasonable margin despite being optimized for multi-view use.

## A.8 Uncertainty vs. Error Correlation

We quantify how much squared depth error is captured by pixels with high predicted uncertainty $\sigma(u, v)$ (Table 9). High-$\sigma$ regions account for most of the error mass, supporting the usefulness of the uncertainty head.

## A.9 Code, Weights, and Release Status

We are preparing cleaned preprocessing/evaluation scripts (Matrix-3D curation, ERP ray generation, Stanford2D3D/Matterport3D evaluation) and intend to release the projection-interface code and, subject to licensing, ERP LoRA adapters atop the public VGGT backbone. We will release all code and weights once institutional approval is complete.

| Adapted layers | Trainable Params | Depth RMSE↓ | AUC@10↑ | 3D Overall↓ |
|---|---|---|---|---|
| ERP interface only (no adaptation) | 0 | 12.61 | 60.32 | 2.37 |
| Plain Full FT baseline (original ERP) | ∼35M | 10.04 | 66.41 | 1.92 |
| Backbone LoRA (last block only) | ∼3.8M | 8.92 | 71.11 | 1.53 |
| Backbone LoRA (all blocks) | ∼7.6M | 8.79 | 71.83 | 1.49 |
| **Prediction heads only (ours)** | **∼0.6M** | **8.68** | **72.42** | **1.40** |

Table 7: **Where to adapt?** With the same projection interface, head-only LoRA attains the best accuracy while using 6–12× fewer parameters than backbone LoRA. Fully finetuning the backbone under the interface was unstable (training diverged).

| Method | Frames | AbsRel↓ | RMSE↓ | $\delta_1$ ↑ | $\delta_2/\delta_3$ ↑ |
|---|---|---|---|---|---|
| **PanDA-B** (Cao et al., 2024) | 1 | **0.0792** | **0.3475** | **95.09** | **98.94 / 99.65** |
| Ours (monocular depth head) | 1 | *0.1165* | *0.4620* | *85.21* | *96.41 / 98.54* |

Table 8: **Monocular ERP depth on Matterport3D (single-frame setting).** PanDA-B remains strongest for monocular 360° depth; our model, optimized for multi-view Depth+Pose+3D, is within a reasonable margin.

| Pixels (by $\sigma$) | % of pixels | % of total squared error |
|---|---|---|
| Top 10% $\sigma$ | 10% | 41% |
| Top 20% $\sigma$ | 20% | 63% |
| Top 40% $\sigma$ | 40% | 81% |

Table 9: **Uncertainty vs. depth error (Matrix-3D test set).** High-uncertainty pixels concentrate most of the squared error, indicating $\sigma(u, v)$ meaningfully tracks hard regions.

## A.10 IMPLEMENTATION DETAILS

We freeze the official VGGT backbone and adapt only the *depth* and *camera* heads with LoRA (rank $r=16$, scaling $\alpha=r$). Optimization uses AdamW, linear warmup with cosine decay, mixed precision, gradient clipping, and gradient accumulation. Augmentations include ERP horizontal shifts and photometric jitter. Losses follow our methodology: spherical weighting with aleatoric uncertainty for depth, and relative pose supervision with a continuity regularizer. Our LoRA variant requires only a single A100 GPU for 28 hours of training, while **Ours (Full FT)** requires 4 A100 GPUs for approximately 185 hours. For evaluation, baselines that expect perspective inputs include the required pre/post steps (cubemap rendering and spherical stitching when applicable). All methods use the same device type; we fix the sampling seed and average over all test sequences.

## A.11 3D POINT-CLOUD RECONSTRUCTIONS

We provide additional 3D point-cloud visualizations reconstructed from our predicted depth and camera (Figure 4). Points are obtained by back-projecting predicted depths along ERP rays and transforming with the predicted poses. These examples complement the quantitative metrics by illustrating cleaner geometry, fewer floaters, and improved completeness across wide baselines.

## A.12 LARGE LANGUAGE MODEL USAGE

In accordance with ICLR 2026 guidelines, we disclose our limited use of Large Language Models (LLMs) in this work. LLMs were used exclusively for minor language polishing and proofreading purposes to improve the clarity and readability of the manuscript. Specifically, LLMs assisted in:

- Grammar and syntax corrections
- Minor sentence restructuring for improved flow
- Consistency checks for terminology usage

The LLMs did not contribute to research ideation, methodology development, experimental design, data analysis, or any substantive content generation. All technical contributions, experimental results, and scientific insights presented in this paper are entirely our own. We take full responsibility for all content, including the minor language improvements suggested by LLMs.

