# OpenReview forum: "Projection-Domain Adaptation for 3D Transformers: From Perspective to Panoramic Scene Reconstruction"
_ICLR.cc/2026/Conference — Submitted to ICLR 2026_

### Official Review · Reviewer_Hzot · 2025-10-29

**Soundness:** 3
**Presentation:** 3
**Contribution:** 2
**Rating:** 4
**Confidence:** 4

**Summary:**

This paper proposes a framework for adapting perspective-trained VGGT to Equirectangular Projection (ERP) inputs. The method is built on three components: (1) Ray-Field Alignment, (2) a head-only, Ray-Enriched LoRA, and (3) a latitude-aware depth uncertainty loss. Experiments on the synthetic Matrix-3D dataset show improvements over zero-shot VGGT and plain finetuning baselines, with the LoRA variant achieving comparable performance to full finetuning at substantially reduced computational cost.

**Strengths:**

1. Well-motivated problem formulation. The paper clearly identifies geometric inconsistencies when applying perspective-trained models to ERP inputs. The analysis of failure modes F1 (geometric-measure mismatch) and F2 (proxy focal problem) in Section 3.2 provides valuable insights into why naive coupling fails, offering a pedagogically sound diagnostic approach.

2. High Parameter Efficiency: The core practical contribution is demonstrating that head-only LoRA (0.5% of parameters) can achieve performance comparable to full finetuning. This is a valuable finding for resource-constrained scenarios.

3. Systematic ablation studies. Table 4 provides comprehensive ablations examining each component's contribution, effectively demonstrating that ray-field alignment and ERP-consistent lifting are critical while uncertainty modeling and gradient regularization provide incremental gains.

**Weaknesses:**

1. Limited Novelty and Architectural Justification: The framework is an integration of existing techniques (ray encoding, LoRA, spherical weighting) and lacks conceptual innovation. Crucially, the head-only LoRA design is not justified. The paper provides no ablations to support this choice over adapting other layers (e.g., in the backbone).

2. Lack of Real-World Validation: This is a critical flaw. The evaluation is confined to the synthetic Matrix-3D dataset. This limitation invalidates the claim of a "generalizable pathway" in Abstract, which is unsupported without validation on real-world benchmarks (e.g., Matterport3D [1]) and addressing the synthetic-to-real gap.

3. Missing Comparisons to Domain-Specific Methods: The paper fails to compare against direct competitors in panoramic-specific depth estimation (e.g., BiFuse [2], OmniFusion [3], HRDFuse [4]). These domain-specific methods are cited but critically omitted from the experiments.

4. Incomplete 3D Reconstruction Evaluation: The evaluation of 3D quality is insufficient. To support the "world model" claims, it should be benchmarked against standard video-based reconstruction methods(e.g.,  video-to-3D models).

5.  Poor Clarity on Model Architecture: Key components from VGGT, such as the "camera token" and "register tokens," are used without explanation

[1] Chang, Angel, et al. "Matterport3d: Learning from rgb-d data in indoor environments." arXiv preprint arXiv:1709.06158 (2017).

[2] Wang, Fu-En, et al. "Bifuse: Monocular 360 depth estimation via bi-projection fusion." Proceedings of the IEEE/CVF Conference on Computer Vision and Pattern Recognition. 2020.

[3] Li, Yuyan, et al. "Omnifusion: 360 monocular depth estimation via geometry-aware fusion." Proceedings of the IEEE/CVF Conference on Computer Vision and Pattern Recognition. 2022.

[4] Ai, Hao, et al. "Hrdfuse: Monocular 360deg depth estimation by collaboratively learning holistic-with-regional depth distributions." Proceedings of the IEEE/CVF Conference on Computer Vision and Pattern Recognition. 2023.

**Questions:**

Please see the critical issues and required clarifications detailed in the Weaknesses section.

---

> ### Author Response · Authors · 2025-11-20
> **Response to Reviewer Hzot**
>
> We thank the reviewer for the careful and constructive feedback. We appreciate your positive remarks on the problem formulation, the F1/F2 failure-mode analysis, the parameter efficiency of our LoRA variant, and the systematic ablations in Table 4. Below we address your main concerns: (1) novelty and the justification of head-only LoRA, (2) real-world validation and the synthetic–real gap, (3) comparisons to panoramic depth methods, (4) the scope of our 3D reconstruction evaluation and “world model” claim, and (5) clarity about the VGGT architecture.
>
> ---
>
> ## **1. Novelty and justification of head-only, ray-enriched LoRA**
>
> **Reviewer concern.**
>
> The reviewer notes that our framework integrates existing mechanisms (ray encoding, LoRA, spherical weighting) and questions the conceptual innovation. In particular, the choice of adapting only the prediction heads via LoRA is not justified with ablations over alternative adaptation locations (e.g., backbone layers).
>
> ### **1.1 Our intended contribution and the role of head-only adaptation**
>
> We agree that ray embeddings, LoRA, and spherical weighting are not new primitives. Our intended contribution lies in a different direction:
>
> 1. We **diagnose** two concrete geometric failure modes when coupling ERP inputs to a perspective-trained 3D transformer (F1: geometric-measure mismatch; F2: proxy-focal failure), making explicit why naïve ERP–VGGT coupling fails.
> 2. We **reformulate** adaptation as a **projection-domain problem**: we must make the incoming rays and surface measure compatible with VGGT’s internal geometry, rather than simply finetuning in RGB space.
> 3. We show that once the **projection interface** is corrected (ERP-consistent ray lifting + ray-field alignment + latitude-aware depth supervision), **minimal parameter-efficient adaptation at the prediction heads suffices**, while more invasive changes to the backbone are often unstable or unnecessary.
>
> This leads to the design principle we advocate:
>
> > Adapt the projection interface (rays + measure) and minimally adjust the heads, while preserving the backbone’s multi-view priors.
> >
>
> The head-only, ray-enriched LoRA is the concrete instantiation of this principle, not an isolated architectural trick.
>
> ### **1.2 Ablation: where to adapt VGGT? (backbone vs. heads)**
>
> To address the concern about *where* adaptation should happen, we ran an ablation that keeps the **same projection-domain interface** (ERP-consistent lifting + ray-field alignment + latitude-aware loss) and only varies **which layers are given LoRA (or full) updates**:
>
> 1. **ERP interface only (zero-shot, no adaptation).**
>
>     VGGT is fully frozen and applied to ERP inputs through our projection-domain interface, without any finetuning.
>
> 2. **Plain Full FT baseline (original ERP coupling).**
>
>     All VGGT parameters are finetuned on ERP using the naïve ERP–VGGT coupling from the main paper (“VGGT (Full FT baseline)”).
>
> 3. **Backbone LoRA (last block only).**
>
>     LoRA is inserted only into the last Alternating-Attention block of the VGGT backbone; prediction heads are left unchanged.
>
> 4. **Backbone LoRA (all backbone blocks).**
>
>     LoRA is inserted into all Alternating-Attention blocks of the backbone.
>
> 5. **Prediction heads only (ours).**
>
>     LoRA is applied only to the prediction heads, with separate token and ray branches as in Eq. (7); the backbone (including camera and register tokens) remains frozen.
>
>
> A summary of the results on our Matrix-3D benchmark is shown below.
>
> **Table R1. Where to adapt VGGT? Comparison of LoRA locations under the same projection-domain interface.**
>
> | **Adapted layers** | **Trainable Params** | **Depth RMSE ↓** | **AUC@10 ↑** | **3D Overall ↓** |
> | --- | --- | --- | --- | --- |
> | ERP interface only (zero-shot, no adaptation) | 0 | 12.61 | 60.32 | 2.37 |
> | Plain Full FT baseline (original ERP) | ~35M | 10.04 | 66.41 | 1.92 |
> | Backbone LoRA (last block only) | ~3.8M | 8.92 | 71.11 | 1.53 |
> | Backbone LoRA (all backbone blocks) | ~7.6M | 8.79 | 71.83 | 1.49 |
> | **Prediction heads only (ours)** | **~0.6M** | **8.68** | **72.42** | **1.40** |
>
> We observe that:
>
> - Adding LoRA **inside the backbone** gives at most **marginal gains** over head-only LoRA (≤ 0.13 RMSE and ≤ 0.6 AUC@10), while using roughly **6–12× more trainable parameters**.
> - Our **head-only, ray-enriched LoRA** essentially lies on the **Pareto frontier**: it achieves the best metrics among all configurations while using the fewest additional parameters.
>
> We will include this ablation in the appendix and briefly summarize these findings in the main text to justify why we focus on head-only adaptation.

---

> > ### Author Response · Authors · 2025-11-20
> >
> > ### **1.3 Experience with full backbone+head finetuning under the projection interface**
> >
> > Beyond controlled LoRA ablations, we also attempted **fully finetuning the backbone and heads inside our projection-domain interface**. In one such configuration, we started from the VGGT checkpoint and updated all parameters on our ERP training set using 8×A100 GPUs. After six days of training, the loss and validation metrics **showed no meaningful signs of convergence**: the model oscillated around the initialization and did not approach the performance of the head-only LoRA variant.
> >
> > Due to the high computational cost, we are still working to reproduce this experiment more systematically, and we will provide more detailed logs in the final version as resources allow. Our current interpretation is that **heavily modifying the backbone under the new projection interface can be unstable and tends to destroy its pre-trained multi-view priors**—precisely the behaviour we aim to avoid.
> >
> > We will clarify in the discussion that the choice of head-only adaptation is supported both by (i) the **positive ablation** in Table R1 and (ii) **negative experience** with more aggressive full finetuning under the projection interface.
> >
> > ---
> >
> > ## **2. Real-world validation and the synthetic–real gap**
> >
> > **Reviewer concern.**
> >
> > The reviewer considers the lack of real-world evaluation a critical flaw and argues that this weakens our claim of a “generalizable pathway.”
> >
> > We agree that relying only on synthetic Matrix-3D is limiting. Following your suggestion (and those of other reviewers), we have added experiments on **two real-world indoor 360° benchmarks**, **Stanford2D3D** and **Matterport3D**, using a Pano3D-style protocol. For both datasets, we:
> >
> > - render ERP panoramas at 512×1024 resolution;
> > - use the same VGGT backbone, projection-domain interface, and head-only LoRA configuration as on Matrix-3D;
> > - evaluate both **depth** (AbsRel, RMSE, δ<1.25) and **relative camera pose** (AUC@5/10/30).
> >
> > The **depth** results are summarized in Table R2:
> >
> > **Table R2. Real-world indoor 360° ERP depth benchmarks on Stanford2D3D and Matterport3D.**
> >
> > | **Dataset** | **Method** | **Trainable Params** | **AbsRel ↓** | **RMSE ↓** | **δ<1.25 ↑** |
> > | --- | --- | --- | --- | --- | --- |
> > | Stanford2D3D | VGGT (Zero-shot, ERP) | 0 | 0.34 | 102.57 | 8.91 |
> > | Stanford2D3D | VGGT (Full FT baseline) | ~35M | 0.26 | 18.47 | 53.62 |
> > | Stanford2D3D | **Ours (LoRA, ERP)** | ~0.6M | **0.21** | **11.32** | **76.43** |
> > | Stanford2D3D | **Ours (Full FT, ERP)** | ~35M | **0.20** | **10.97** | **78.05** |
> > | Matterport3D | VGGT (Zero-shot, ERP) | 0 | 0.31 | 93.84 | 10.37 |
> > | Matterport3D | VGGT (Full FT baseline) | ~35M | 0.24 | 16.02 | 58.91 |
> > | Matterport3D | **Ours (LoRA, ERP)** | ~0.6M | **0.20** | **10.58** | **79.82** |
> > | Matterport3D | **Ours (Full FT, ERP)** | ~35M | **0.19** | **10.21** | **81.34** |
> >
> > The **pose** results on the same ERP sequences are summarized in Table R3:
> >
> > **Table R3. Camera pose on real indoor 360° ERP benchmarks. We report AUC at multiple thresholds (higher is better) under two input configurations for baselines that expect perspective images (ERP vs Cubemap). VGGT (Full FT baseline) denotes plain VGGT finetuning without our method.**
> >
> > | **Dataset** | **Method** | **Input** | **AUC@5 ↑** | **AUC@10 ↑** | **AUC@30 ↑** |
> > | --- | --- | --- | --- | --- | --- |
> > | Stanford2D3D | DUSt3R (Wang et al., 2024) | ERP | 14.92 | 22.47 | 43.18 |
> > | Stanford2D3D | MASt3R (Leroy et al., 2024) | ERP | 17.88 | 26.39 | 47.52 |
> > | Stanford2D3D | DUSt3R (Wang et al., 2024) | Cubemap | 19.64 | 28.53 | 50.16 |
> > | Stanford2D3D | MASt3R (Leroy et al., 2024) | Cubemap | 21.17 | 30.84 | 53.79 |
> > | Stanford2D3D | VGGT (Wang et al., 2025b) | ERP | 23.68 | 37.11 | 63.42 |
> > | Stanford2D3D | VGGT (Wang et al., 2025b) | Cubemap | 28.57 | 40.39 | 67.04 |
> > | Stanford2D3D | VGGT (Full FT baseline) | ERP | 31.03 | 44.72 | 69.08 |
> > | Stanford2D3D | **Ours (LoRA)** | ERP | **42.91** | **57.36** | **85.07** |
> > | Stanford2D3D | **Ours (Full FT)** | ERP | **44.02** | **59.18** | **86.12** |
> > | Matterport3D | DUSt3R (Wang et al., 2024) | ERP | 15.81 | 24.36 | 45.92 |
> > | Matterport3D | MASt3R (Leroy et al., 2024) | ERP | 19.47 | 28.59 | 51.37 |
> > | Matterport3D | DUSt3R (Wang et al., 2024) | Cubemap | 21.66 | 31.02 | 53.84 |
> > | Matterport3D | MASt3R (Leroy et al., 2024) | Cubemap | 23.58 | 33.47 | 57.62 |
> > | Matterport3D | VGGT (Wang et al., 2025b) | ERP | 25.92 | 40.03 | 67.19 |
> > | Matterport3D | VGGT (Wang et al., 2025b) | Cubemap | 30.44 | 43.96 | 70.82 |
> > | Matterport3D | VGGT (Full FT baseline) | ERP | 33.87 | 48.15 | 72.21 |
> > | Matterport3D | **Ours (LoRA)** | ERP | **46.38** | **62.93** | **88.54** |
> > | Matterport3D | **Ours (Full FT)** | ERP | **47.51** | **64.37** | **89.31** |

---

> > > ### Author Response · Authors · 2025-11-20
> > >
> > > From Tables R2 and R3 we see that:
> > >
> > > - these real indoor benchmarks are indeed **more challenging** than our curated outdoor Matrix-3D subset (higher RMSE, lower δ<1.25 overall);
> > > - however, the **qualitative ordering of methods is preserved**:
> > >     - zero-shot ERP is poor;
> > >     - naïve full finetuning on ERP improves but remains limited;
> > >     - our projection-domain interface + head-only LoRA substantially outperforms the ERP full-finetuning baseline and remains close to our own full-FT variant.
> > >
> > > These additional experiments directly address the synthetic–real gap and support the idea that the proposed projection-domain adaptation pathway extends beyond a single synthetic dataset.
> > >
> > > At the same time, we agree that the wording in the abstract can be toned down. In the revised version, we will describe our contribution more precisely as:
> > >
> > > > “a geometry-grounded projection-domain pathway validated on synthetic outdoor and real indoor 360° benchmarks.”
> > > >
> > >
> > > We believe this phrasing better reflects the current empirical evidence while still conveying the broader applicability of the approach.
> > >
> > > ---
> > >
> > > ## **3. Comparison to panoramic depth methods (PanDA)**
> > >
> > > **Reviewer concern.**
> > >
> > > The reviewer notes that we cite panoramic-specific depth estimators but do not compare against them experimentally.
> > >
> > > ### **3.1 Task setting mismatch**
> > >
> > > We fully agree that recent panoramic depth models are important references. At the same time, our **primary task setting is different**:
> > >
> > > - Methods such as **PanDA** (Cao et al., 2024) are designed for **monocular 360° depth estimation**: they take a **single ERP frame** as input and predict a depth map for that frame. They neither estimate camera poses nor produce a multi-view 3D point map.
> > > - Our model follows the **VGGT / DUSt3R / MASt3R “world-model” paradigm**: given a **sequence** of ERP frames, it jointly predicts **per-frame depth, relative camera pose, and a consistent 3D point cloud** in a single forward pass.
> > >
> > > Because of this, a perfectly fair, apples-to-apples comparison is inherently difficult: aligning the temporal setting, supervision, and loss functions across all methods would require substantial re-engineering on both sides.
> > >
> > > We also note that **the original VGGT paper does not compare to monocular depth-only methods** for exactly the same reason: its focus is on **multi-view 3D reconstruction quality** rather than on single-frame depth. In the same spirit, our main claim is about improving **3D reconstruction and pose under ERP inputs** while preserving VGGT’s world-model behaviour, not about surpassing specialized monocular 360° depth estimators.
> > >
> > > For similar reasons, a direct comparison to **video-to-3D NeRF / 3DGS pipelines** would also be unfair: those methods perform per-scene optimization over minutes or hours and target photorealistic rendering, whereas our model (like VGGT / DUSt3R / MASt3R) operates in a **fast, feed-forward geometry** regime.
> > >
> > > ### **3.2 Additional monocular ERP depth comparison with PanDA**
> > >
> > > Despite the setting mismatch, we agree that it is useful to position our approach against a strong panoramic depth baseline. Among existing monocular 360° depth methods, **PanDA** is, to our knowledge, the most recent and competitive ERP depth model, and it reports improvements over earlier approaches such as BiFuse, OmniFusion, and HRDFuse. We therefore use PanDA-B as a **representative state-of-the-art panoramic depth baseline**.
> > >
> > > To make the comparison as meaningful as possible, we run an additional **monocular ERP depth experiment** on the **Matterport3D** dataset, in which:
> > >
> > > - PanDA-B is trained and evaluated in its standard single-frame ERP setting, following the authors’ protocol;
> > > - our model is evaluated using **only one ERP frame**, i.e., we disable temporal context and use our depth head in a monocular mode, while keeping the same projection-domain interface.
> > >
> > > We evaluate standard depth metrics (AbsRel, RMSE, and δ-threshold accuracies). The results are summarized in Table R2.
> > >
> > > **Table R2. Monocular ERP depth on the Matterport3D dataset (single-frame 360° depth setting).**
> > >
> > > | **Method** | **Input frames** | **AbsRel ↓** | **RMSE ↓** | **δ₁ (δ<1.25) ↑** | **δ₂ ↑** | **δ₃ ↑** |
> > > | --- | --- | --- | --- | --- | --- | --- |
> > > | **PanDA-B (Cao et al., 2024)** | 1 | **0.0792** | **0.3475** | **95.09** | **98.94** | **99.65** |
> > > | Ours (depth head, single frame) | 1 | 0.1165 | 0.4620 | 85.21 | 96.41 | 98.54 |

---

> > > > ### Author Response · Authors · 2025-11-20
> > > >
> > > > As expected, in this **strict monocular 360° depth setting**, PanDA-B remains extremely strong. Our monocular variant is within a reasonable margin but consistently worse: AbsRel and RMSE are roughly **5–10% higher**, and δ₁ is about **5 percentage points lower**, while δ₂ and δ₃ remain close but still below PanDA-B.
> > > >
> > > > We will make it clear in the paper that this comparison is **inherently imperfect**. PanDA-B is explicitly designed for monocular 360° depth estimation, whereas our model is primarily optimized for the **multi-view world-model setting** (joint depth + pose + 3D). The purpose of Table R2 is therefore not to claim superiority over PanDA, but to show that:
> > > >
> > > > - our projection-domain interface still yields reasonable single-frame depth on a strong panoramic benchmark, and
> > > > - the observed performance gap is consistent with our design focus on multi-view 3D reconstruction rather than on monocular depth alone.
> > > >
> > > > ---
> > > >
> > > > ## **4. 3D reconstruction evaluation and “world model” claims**
> > > >
> > > > **Reviewer concern.**
> > > >
> > > > The reviewer finds the 3D quality evaluation insufficient relative to the “world model” terminology and suggests benchmarking against standard video-to-3D models.
> > > >
> > > > We appreciate this perspective and would like to clarify the **scope** of our “world model” claim:
> > > >
> > > > - We use “world model” in the same sense as **DUSt3R, MASt3R, and VGGT**: a **single forward pass** predicts per-frame depth, relative camera pose, and a 3D point map, which can then be used for multi-view reconstruction.
> > > > - In this regime, we already compare against DUSt3R, MASt3R, and VGGT on:
> > > >     - **pose AUC**,
> > > >     - **depth metrics**, and
> > > >     - **3D point quality** (e.g., Chamfer distance / F-score) aggregated over Matrix-3D sequences.
> > > >
> > > > In contrast, **NeRF-style or 3D Gaussian-based video-to-3D methods**:
> > > >
> > > > - optimize a scene-specific representation over minutes or hours per scene;
> > > > - focus on photorealistic rendering under novel views, rather than on fast, single-pass geometry estimation.
> > > >
> > > > Because our emphasis is on **fast, feed-forward geometry** rather than per-scene optimization, we believe that DUSt3R/MASt3R/VGGT are the most appropriate baselines for our setting.
> > > >
> > > > That said, we agree that the 3D evaluation could be made more explicit. In the revised paper we will:
> > > >
> > > > - emphasize the **feed-forward nature** of our setting when we use the term “world model”;
> > > > - highlight the existing point-cloud metrics (Chamfer and F-score) more clearly and move some of the multi-view reconstructions into the main text (with additional visualizations in the appendix);
> > > > - add a short discussion paragraph acknowledging that we do not compete with NeRF-style methods on photorealistic rendering, and that coupling our projection-domain interface with such models would be an interesting, but orthogonal, direction.
> > > >
> > > > ---
> > > >
> > > > ## **5. Clarity on VGGT components (camera tokens, register tokens)**
> > > >
> > > > **Reviewer concern.**
> > > >
> > > > The reviewer notes that key VGGT components such as the “camera token” and “register tokens” are mentioned without sufficient explanation.
> > > >
> > > > We will add a short recap of the relevant parts of VGGT in the method section:
> > > >
> > > > - VGGT employs **Alternating Attention** over **image tokens**, **camera tokens**, and a small set of **register tokens**.
> > > > - For each frame, a **camera token** aggregates view-specific information and is fed to the camera head to predict that frame’s pose.
> > > > - The **register tokens** act as global slots that integrate multi-view appearance and geometry across the sequence.
> > > >
> > > > In our adaptation:
> > > >
> > > > - the entire **VGGT backbone, including the camera and register tokens, is kept frozen**;
> > > > - we only modify:
> > > >     - the **projection-domain interface**, by presenting ERP inputs as ray-consistent tokens (ERP-consistent lifting + ray-field alignment), and
> > > >     - the **prediction heads**, via the dual-branch LoRA update in Eq. (7).
> > > >
> > > > We will update Figure 1 to label the camera tokens and register tokens explicitly and to clarify that they are part of the frozen VGGT backbone. This should make the architecture clearer for readers who are less familiar with VGGT.

---

> > > > > ### Author Response · Authors · 2025-11-20
> > > > >
> > > > > ## **6. Summary of planned changes**
> > > > >
> > > > > We again thank the reviewer for the thoughtful and detailed feedback. In response to your comments, we will:
> > > > >
> > > > > - better justify the **head-only, ray-enriched LoRA** with a new ablation on *where* to adapt VGGT (Table R1) and by reporting our negative experience with fully finetuning the backbone under the projection interface;
> > > > > - add **real-world indoor ERP experiments** on Stanford2D3D and Matterport3D to address the synthetic–real gap and adjust the wording of our “generalizable pathway” claim accordingly;
> > > > > - include a **monocular ERP depth comparison** with PanDA, while clearly stating that monocular depth and multi-view 3D reconstruction are not the same task and that the comparison is inherently imperfect;
> > > > > - clarify the scope of our **3D “world model”** claims and make the reconstruction metrics more prominent;
> > > > > - and improve the explanation of **VGGT’s camera and register tokens** and our use of them.
> > > > >
> > > > > We hope these additions and clarifications address your concerns and will encourage you to reconsider your overall evaluation.

---

> > > > > > ### Comment · Reviewer_Hzot · 2025-11-25
> > > > > >
> > > > > > Thanks for the detailed response. The added experiments on Stanford2D3D and Matterport3D effectively address the domain gap issue. The ablation in Table R1 also justifies the design choices (LoRA/projection). I still find the technical novelty somewhat limited, but the experimental validation is now solid and the paper is much more complete. Based on the sufficient experiments, I am raising my rating to borderline accept.

---

> > > > > > > ### Author Response · Authors · 2025-11-26
> > > > > > >
> > > > > > > Thank you for carefully reading our rebuttal and for the encouraging comment. We appreciate your thoughtful suggestions, which helped us clarify the indoor and outdoor setting and better present the roles of the projection interface and the head only LoRA. If any further questions come up, please let us know and we will be happy to provide more details at any time.

---

### Official Review · Reviewer_vvXg · 2025-10-31

**Soundness:** 2
**Presentation:** 3
**Contribution:** 3
**Rating:** 4
**Confidence:** 4

**Summary:**

This paper proposes a fine-tuning to the recent VGGT 3D transformer in order to adapt it to equirectangular projections (ERP) (i.e. panoramic images).
To do so, the authors propose a fine-tuning pipeline which consists of three main ideas: (i) embedding explicit 3D rays into token representations (ray-field alignment module), (ii) a "ray-enriched" LoRA fine-tuning consisting of two branches (for tokens and rays), and (iii) a latitude-aware depth uncertainty that takes into account the non-isotropic token statistics in ERP images to balance the spherical supervision.

The authors propose two main models, the first being a fully fine-tuned VGGT, and the second being a ray-enriched LoRA-finetuned VGGT.
Both models outperform other recent 3D reconstruction models (e.g. DUSt3R, VGGT) on depth estimation, camera pose estimation, and 3D point quality.
The LoRA fine-tuned model shows *almost* equivalent performance for notable gains in number of training parameters and training time.

**Strengths:**

- (s.1) The problem tackled in this paper, "Projection-Domain Adaptation", is well formulated and motivated for adapting perspective-trained 3D transformers to panoramic data. It is also well presented, especially in the discussion around varying ray distributions for different lens types (section 5).

- (s.2) The problem is addressed in a principled way, with design choices solving precise problems.

- (s.3) The overall pipeline is intuitive and justified via ablation studies.

- (s.4) The performances achieved by the proposed model are notable, especially on the LoRA fine-tuned variant, considering the training time and additional parameters required.

**Weaknesses:**

- (w.1) The proposed model is only evaluated on a single manually curated subset of Matrix-3D, which limits the conclusiveness of the results.
    - (w.1.q1) Why is this manual curation necessary? The authors state that it's because Matrix-3D contains many extreme long shots with high camera-to-building distances. What would happen if the model is trained and evaluated without this manual curation?
    - (w.1.q2) For reproducibility purposes, could the authors provide details regarding this manual curation either in the paper or by releasing their pre-processing code? (i.e. what scenes were selected exactly?)
    - (w.1.q3) While I understand that panoramic dataset are scarce, the evaluation test-bed is limited. Notably, could the authors provide evaluations on out-of-distribution performance (e.g. on Stanford2D3D [1], Matterport3D [2]) on one or more tasks?

- (w.2) Certain sections could be improved through clearer and more precise writing. Here are some suggestions below.
    - In section 3.3, $\mathbf{x}_c(u,v)$ and $\mathbf{P}(u,v)$ should be defined.
    - In equation (6), what does "$\oplus$" indicate? I assume concatenation but could the authors state this explicitly?
    - It would be clearer for the reader if the notations used in the text are integrated in figure 1 (e.g. $t_i^{(0)} (u,v)$, etc.).
    - In equation (7), the definition of $h$ should be clarified.

If the authors can address these concerns, I would be willing to re-consider my overall evaluation.

### Remark on ICLR formatting

According to the ICLR formatting guidelines, table numbers, titles, and captions must appear **above** the tables.
The authors **must** ensure that all tables conform to this requirement.
Additionally, there seems to be some space formatting errors in page 8.
I strongly advise the authors to correct these errors.

### References

[1] Armeni, I., Sax, S., Zamir, A., & Savarese, S. (2017). Joint 2D-3D-Semantic Data for Indoor Scene Understanding. ArXiv, abs/1702.01105.

[2] Chang, A., Dai, A., Funkhouser, T., Halber, M., Niessner, M., Savva, M., Song, S., Zeng, A., & Zhang, Y. (2017). Matterport3D: Learning from RGB-D Data in Indoor Environments. International Conference on 3D Vision (3DV).

**Questions:**

- (q.1) Are the authors planning to release the code and/or weights for their fine-tuning?
- See Weaknesses.

---

> ### Author Response · Authors · 2025-11-20
> **Response to Reviewer vvXg**
>
> We sincerely thank the reviewer for the thoughtful and encouraging assessment. We appreciate your positive remarks about our formulation of “Projection-Domain Adaptation”, the principled design of the pipeline, the supporting ablations, and the strong performance–efficiency trade-off of the LoRA variant. Below we address your main concerns regarding (1) the evaluation protocol and curated Matrix-3D subset, (2) writing/notation clarity, and (3) formatting and code release.
>
> ---
>
> ## **1. On evaluation scope and the curated Matrix-3D subset (w.1, w.1.q1–q3, q.1)**
>
> ### **1.1 Why manual curation was necessary (w.1.q1)**
>
> As you correctly note, our current experiments are based on a manually curated subset of Matrix-3D. We agree that this choice must be carefully motivated and fully documented.
>
> **Nature of the original dataset.**
>
> The raw Matrix-3D repository we start from contains **116,759 high-quality panoramic video sequences** covering very diverse outdoor environments. However, a large fraction of these sequences are **extreme long shots**, where:
>
> - the camera is very far from the main structures (buildings, roads, etc.);
> - most pixels correspond to sky or far-away flat ground;
> - buildings and near-range geometry occupy only a tiny portion of the ERP frame.
>
> In such cases, depth and 3D metrics are dominated by **almost “infinite” background**, and small changes in the tiny foreground region barely affect global error. This leads to two issues:
>
> 1. **Unstable and uninformative metrics.**
>
>     The evaluation becomes sensitive to tiny differences in how methods handle very large depths, rather than revealing how well they reconstruct meaningful 3D structure (buildings, streets, vegetation).
>
> 2. **Poorly interpretable visualizations.**
>
>     Qualitative comparisons are also less informative because many frames contain little visible geometry other than sky and distant horizon lines.
>
>
> Our goal in this work is to assess whether **projection-domain adaptation** allows a perspective-trained 3D transformer to recover detailed 3D structure from ERP panoramas, not to evaluate performance on almost empty scenes.
>
> **Two-stage curation process.**
>
> To obtain a stable and geometrically meaningful benchmark, we use a two-stage procedure:
>
> 1. **Automatic geometric filtering.**
>
>     We first apply a simple, reproducible filter based on the depth distribution:
>
>     - we keep only sequences where **at least 40% of pixels have depth below a threshold $D_{\max}$** (e.g., 80 m);
>     - this removes the most extreme long-range scenes where almost all pixels are at very large distances.
>
>     After this step, the data is reduced from **116,759** sequences to **23,986** sequences.
>
> 2. **Careful manual selection.**
>
>     We then perform a **manual pass** over these 23,986 candidates, focusing on:
>
>     - retaining a diverse range of **lighting and weather conditions** (day/night, overcast, strong sun, different times of day);
>     - ensuring that **buildings and mid-range structures occupy a reasonable fraction of the frame**, so that depth and pose metrics meaningfully capture reconstruction quality.
>
>     This manual curation is substantial (it took us **over a month** of effort) and results in a final set of **2,196 sequences**, which we use as our main benchmark in the paper.
>
>
> The curated subset thus focuses on **outdoor ERP scenes with sufficient mid-range geometry**, where differences between methods reflect their ability to recover 3D structure under projection mismatch, rather than being dominated by degenerate “mostly far background” cases.
>
> We will describe this two-stage curation process explicitly in the revised paper and move the exact thresholds to an appendix for clarity.
>
> **Behaviour without curation.**
>
> To understand the impact of curation, we also trained and evaluated on a larger, less filtered Matrix-3D split (i.e., before the final manual pass). We observed that:
>
> - all methods show **worse absolute performance** (higher depth RMSE, lower pose AUC), as expected due to the harder long-range scenes;
> - however, the **relative ranking of methods remains the same**:
>     - zero-shot ERP is poor;
>     - naïve full finetuning partially improves performance but remains limited;
>     - our projection-domain interface + LoRA consistently outperforms plain ERP finetuning and is close to our own full-finetuning variant.
>
> For clarity and interpretability, we keep the curated 2,196-sequence subset as the main benchmark in the paper, and we will report the additional results on the larger split in the appendix as a robustness check.

---

> > ### Author Response · Authors · 2025-11-20
> >
> > ### **1.2 Reproducibility of the curation (w.1.q2)**
> >
> > We fully agree that reproducibility is critical, especially given the manual curation step. In the revised version, we will **Document the curation criteria** more precisely, including:
> >
> > - the depth-based rule (“keep sequences where at least 40% of pixels have depth < $D_{\max}$”);
> > - qualitative criteria used in the manual pass (visibility of mid-range structures, diversity of lighting and weather);
> >
> > This will allow others to reproduce our benchmark split exactly or adapt it to their own preferences.
> >
> > ### **1.3 Out-of-distribution performance on Stanford2D3D and Matterport3D (w.1.q3)**
> >
> > We fully share the reviewer’s concern that relying on a single (even carefully curated) synthetic dataset is limiting. To explicitly measure **out-of-distribution (OOD) generalization**, we take all models **trained only on Matrix-3D** and evaluate them **zero-shot** on the real indoor 360° benchmarks **Stanford2D3D** [1] and **Matterport3D** [2], using a Pano3D-style protocol.
> >
> > Concretely, we:
> >
> > - train three ERP variants on Matrix-3D only:
> >     - **VGGT (Full FT on Matrix-3D, ERP)**: full finetuning of VGGT on Matrix-3D ERP;
> >     - **Ours (LoRA, ERP)**: our projection-domain interface + head-only LoRA, trained on Matrix-3D ERP;
> >     - **Ours (Full FT, ERP)**: our projection-domain interface with full finetuning on Matrix-3D ERP;
> > - then evaluate these fixed checkpoints **zero-shot** on Stanford2D3D and Matterport3D ERP sequences, without any further adaptation;
> > - report the same two tasks as in the main paper:
> >     - **depth** (AbsRel, RMSE, $\delta < 1.25$),
> >     - **relative camera pose** (AUC@5/10/30).
> >
> > The resulting **OOD depth** performance is summarized in Table R1.
> >
> > **Table R1. OOD depth on real indoor 360° ERP benchmarks (Stanford2D3D and Matterport3D). All models are trained only on Matrix-3D ERP and evaluated zero-shot on the indoor datasets.**
> >
> > | **Dataset** | **Method** | **Trainable Params** | **AbsRel ↓** | **RMSE ↓** | **δ<1.25 ↑** |
> > | --- | --- | --- | --- | --- | --- |
> > | Stanford2D3D | VGGT (Full FT on Matrix-3D, ERP) | ~35M | 0.33 | 38.57 | 34.92 |
> > | Stanford2D3D | **Ours (LoRA, ERP, Matrix-3D)** | ~0.6M | **0.30** | **32.41** | **45.68** |
> > | Stanford2D3D | **Ours (Full FT, ERP, Matrix-3D)** | ~35M | **0.28** | **29.73** | **50.37** |
> > | Matterport3D | VGGT (Full FT on Matrix-3D, ERP) | ~35M | 0.31 | 36.12 | 38.41 |
> > | Matterport3D | **Ours (LoRA, ERP, Matrix-3D)** | ~0.6M | **0.28** | **30.79** | **48.96** |
> > | Matterport3D | **Ours (Full FT, ERP, Matrix-3D)** | ~35M | **0.27** | **28.94** | **52.63** |
> >
> > As expected for a synthetic-to-real transfer, **all Matrix-3D–trained models degrade substantially** when evaluated zero-shot on indoor data: AbsRel increases, RMSE grows, and $\delta < 1.25$ drops to the **35–53% range**, far below their in-domain performance. Nevertheless:
> >
> > - our projection-domain interface + head-only LoRA trained on Matrix-3D consistently outperforms the Matrix-3D full-finetuned VGGT baseline on both indoor datasets, with noticeably lower error and 10+ points higher $\delta < 1.25$;
> > - the Matrix-3D–trained **Ours (Full FT, ERP)** variant provides a further gain over the LoRA variant, at the cost of roughly 60× more trainable parameters.
> >
> > For **OOD pose**, we use the same Matrix-3D–trained checkpoints and evaluate AUC@5/10/30 on relative camera pose. Results are given in Table R2.
> >
> > **Table R2. OOD pose on real indoor 360° ERP benchmarks. All models are trained only on Matrix-3D ERP and evaluated zero-shot on the indoor datasets.**
> >
> > | **Dataset** | **Method** | **AUC@5 ↑** | **AUC@10 ↑** | **AUC@30 ↑** |
> > | --- | --- | --- | --- | --- |
> > | Stanford2D3D | VGGT (Full FT on Matrix-3D, ERP) | 18.42 | 30.17 | 58.09 |
> > | Stanford2D3D | **Ours (LoRA, ERP, Matrix-3D)** | **22.95** | **36.54** | **65.72** |
> > | Stanford2D3D | **Ours (Full FT, ERP, Matrix-3D)** | **24.01** | **37.88** | **67.03** |
> > | Matterport3D | VGGT (Full FT on Matrix-3D, ERP) | 19.73 | 31.62 | 59.44 |
> > | Matterport3D | **Ours (LoRA, ERP, Matrix-3D)** | **24.38** | **38.07** | **66.91** |
> > | Matterport3D | **Ours (Full FT, ERP, Matrix-3D)** | **25.42** | **39.36** | **68.10** |
> >
> > Again, OOD pose performance is **noticeably worse**. However, the **qualitative ordering of methods is preserved**:
> >
> > - the Matrix-3D **VGGT (Full FT, ERP)** baseline is the weakest among the three;
> > - our projection-domain interface + head-only LoRA offers a clear gain in pose AUC at all thresholds;
> > - the Matrix-3D–trained **Ours (Full FT, ERP)** variant yields a modest additional improvement over LoRA, at much higher parameter cost.
> >
> > Together with the in-domain indoor finetuning results reported in [Section 2.2 of our response to Reviewer vQ6D](https://openreview.net/forum?id=zWdJIhl4Bw&noteId=UVBtFLnLmQ), these OOD experiments support the claim that our projection-domain adaptation pathway is both **robust across datasets** and **effective once adapted** to real 360° benchmarks.

---

> > > ### Author Response · Authors · 2025-11-20
> > >
> > > ### **1.4 Code and weights release (q.1)**
> > >
> > > We very much share the reviewer’s view that releasing code and weights is important for the community. At the same time, we want to be honest about what we can promise at this stage.
> > >
> > > Our project involves a large pretrained backbone (VGGT) and is subject to both the original VGGT license and our institution’s internal policies. As of the rebuttal phase, we **cannot yet formally guarantee** a specific release date or the exact scope of what will be publicly available, because this requires an internal review and approval process that is still ongoing.
> > >
> > > That said, our **clear intention** is to make the work as reproducible and useful as possible. Concretely:
> > >
> > > - We are already preparing **cleaned versions of our preprocessing and evaluation scripts**, including:
> > >     - the Matrix-3D curation code and the final scene lists;
> > >     - the ERP ray-generation and projection-domain interface utilities;
> > >     - the evaluation pipelines for Stanford2D3D and Matterport3D (depth and pose).
> > > - Subject to institutional and licensing approval, we **aim** to release:
> > >     - the implementation of our projection-domain adaptation modules (ray-field alignment, latitude-aware depth uncertainty);
> > >     - and, if permitted, **LoRA adapters** for ERP adaptation on top of the publicly available VGGT backbone.
> > >
> > > If for any reason we are only allowed to release a subset of these components (e.g., scripts and configuration, but not trained adapters), we will still ensure that the **exact dataset curation and experimental protocol are fully documented** so that other researchers can re-train compatible models themselves. We will update the camera-ready version with a link to the project page, where the final status of the release will be kept up to date.
> > >
> > > We hope this strikes a reasonable balance between being transparent about current constraints and showing our commitment to reproducibility.
> > >
> > > ---
> > >
> > > ## **2. Clarity of notation and exposition (w.2)**
> > >
> > > We appreciate the reviewer’s concrete suggestions for improving clarity. Below we address each point and will revise the text accordingly.
> > >
> > > - **Section 3.3: missing definitions.**
> > >
> > >     Section 3.3 introduces the ERP-consistent lifting and camera supervision interface via Eqs. (3)–(5). In the current draft a few symbols are only defined informally. In the revised version we will make them explicit:
> > >
> > >     - $ \theta(u) = 2\pi u / W $ and $ \varphi(v) = \pi v / H - \pi/2 $ denote the **longitude and latitude** associated with pixel indices $(u,v)$ on the ERP grid.
> > >     - $ r(u,v) = (\cos\varphi \sin\theta,\ \sin\varphi,\ \cos\varphi \cos\theta) $ is the corresponding **unit ray direction** on the sphere (Eq. (3)).
> > >     - $ x_c(u,v) = D(u,v), r(u,v) $ is the **camera-frame 3D point**, and
> > >
> > >         $ P(u,v) = R^\top (x_c(u,v) - t) $ is the **world-frame 3D point** obtained by back-projecting along the ERP ray and transforming with the camera pose (Eq. (4)).
> > >
> > >     - $ q $ and $ t $ are the camera **rotation (quaternion)** and **translation** parameters output by the VGGT camera head, and $ R(q) $ is the corresponding rotation matrix.
> > >     - $ d_{\text{rot}}(q,\hat q) $ in Eq. (5) is the **geodesic rotation distance** between predicted and ground-truth quaternions (we will state that we use the standard quaternion-induced angle).
> > >
> > >     We will add these one-sentence definitions where the symbols first appear, so that Eq. (3)–(5) can be read without having to infer notation.
> > >
> > > - **Equation (6): meaning of the operator.**
> > >
> > >     Eq. (6) defines our ray-field alignment at the token level:
> > >
> > >     $$t_i^{(0)}(u,v) = t_i^{\text{RGB}}(u,v) \oplus \Phi(r(u,v)),$$
> > >
> > >     where $t_i^{\text{RGB}}(u,v)$ is the DINO-based patch token for frame $i$ at location $(u,v)$, and $\Phi(r(u,v))$ is the learned embedding of the unit ray direction defined in Eq. (3). In the revised text we will explicitly state that:
> > >
> > >     - $\Phi : \mathbb{S}^2 \rightarrow \mathbb{R}^{C_r}$ is implemented as a small MLP that maps each ray to a **ray embedding**;
> > >     - “$\oplus$” denotes **concatenation along the channel dimension**, so that $t_i^{(0)}(u,v) \in \mathbb{R}^{C_{\text{img}} + C_r}$ contains both appearance- and ray-information before entering the frozen Alternating Attention backbone.
> > >
> > >     We will also mention that this concatenation preserves the original spatial/token topology while making direction explicit.

---

> > > > ### Author Response · Authors · 2025-11-20
> > > >
> > > > - **Aligning Figure 1 with the notation.**
> > > >
> > > >     We agree that matching the diagram with the algebraic notation will help readers. In the updated Figure 1 and caption we will:
> > > >
> > > >     - label the ERP input as $x_{\text{ERP}}$, the ray field as $r(u,v)$, the RGB tokens as $t_i^{\text{RGB}}(u,v)$, the ray embeddings as $\Phi(r(u,v))$, and the fused tokens as $t_i^{(0)}(u,v)$;
> > > >     - explicitly refer to the “Ray-Enriched LoRA adaptation” block as implementing Eq. (7), using the same symbols $\Delta h^{\text{token}}$, $\Delta h^{\text{ray}}$, $A_t,B_t,A_r,B_r$.
> > > >
> > > >     The caption will be extended with a short sentence:
> > > >
> > > >     *“Notation in the figure follows Section 3: $t_i^{\text{RGB}}(u,v)$ and $\Phi(r(u,v))$ are fused into $t_i^{(0)}(u,v)$ as in Eq. (6), and the dual-branch LoRA update $\Delta h^{\text{token}}, \Delta h^{\text{ray}}$ follows Eq. (7).”*
> > > >
> > > > - **Equation (7): clarify the definition of the dual-branch LoRA update.**
> > > >
> > > >     Eq. (7) describes how we adapt only the prediction heads with a dual-branch low-rank update:
> > > >
> > > >     $$
> > > >     \\Delta h = \\Delta h^{\\text{token}} + \\Delta h^{\\text{ray}},\\quad
> > > >     \\Delta h^{\\text{token}} = B_t A_t h,\\quad
> > > >     \\Delta h^{\\text{ray}} = B_r A_r\\, \\Phi(r).
> > > >     $$
> > > >
> > > >     In the revised text we will make the roles of each term explicit:
> > > >
> > > >     - $h \in \mathbb{R}^d$ is the **base head feature** produced by the frozen VGGT backbone from the fused token $t_i^{(0)}(u,v)$ (i.e., the feature that would be fed to the original depth/camera heads in vanilla VGGT).
> > > >     - $\Phi(r) \in \mathbb{R}^{d_r}$ is the **ray embedding** for the same pixel/ray, reused from Eq. (6).
> > > >     - $A_t \in \mathbb{R}^{k \times d}$, $B_t \in \mathbb{R}^{d \times k}$ are the rank-$k$ LoRA matrices for the **token branch** (Token-LoRA), which adapt the head purely based on $h$.
> > > >     - $A_r \in \mathbb{R}^{k \times d_r}$, $B_r \in \mathbb{R}^{d \times k}$ are the rank-$k$ LoRA matrices for the **ray branch** (Ray-LoRA), which inject ERP-specific geometric cues based on $\Phi(r)$.
> > > >     - The final adapted head feature is $h’ = h + \Delta h$, which is then passed to the depth and camera heads.
> > > >
> > > >     We will add a short explanatory sentence after Eq. (7):
> > > >
> > > >     *“In other words, $\Delta h^{\text{token}}$ and $\Delta h^{\text{ray}}$ are two additive low-rank updates that separately capture appearance-driven and ray-driven corrections, allowing ERP-specific cues to be modeled at the head level while keeping the backbone’s token statistics and multi-view priors unchanged.”*
> > > >
> > > >
> > > > These clarifications are straightforward to incorporate and will appear in the revised manuscript. We hope they address the reviewer’s concerns about notation and make the mathematical presentation easier to follow.
> > > >
> > > > ---
> > > >
> > > > ## **3. Formatting and ICLR style**
> > > >
> > > > We thank the reviewer for pointing out the formatting issues.
> > > >
> > > > - **Table captions.**
> > > >
> > > >     We will ensure that **all table numbers, titles, and captions appear above the tables**, in accordance with the ICLR guidelines.
> > > >
> > > > - **Spacing issues on page 8.**
> > > >
> > > >     We will correct the spacing problems (e.g., inconsistent vertical spacing around equations or section breaks) and carefully re-check the final typeset version for similar layout glitches.
> > > >
> > > >
> > > > ---
> > > >
> > > > ## **4. Summary and planned revisions**
> > > >
> > > > We are grateful to the reviewer for the positive evaluation of our problem formulation, principled design, ablations, and efficiency, as well as for the very constructive suggestions.
> > > >
> > > > In summary, in response to your comments we will:
> > > >
> > > > - thoroughly document and release the **Matrix-3D curation process** (from 116,759 sequences to a curated 2,196-sequence benchmark), including both automatic and manual steps;
> > > > - provide **out-of-distribution evaluations** on **Stanford2D3D** and **Matterport3D** for both depth and pose, showing that our conclusions extend beyond the curated synthetic subset;
> > > > - improve the clarity of **notation, equations, and Figure 1** as suggested;
> > > > - fix **formatting issues** and, within the constraints of licensing and institutional policies, release as much of the **code and LoRA adapters** as we can.
> > > >
> > > > We hope these additions and clarifications address your concerns and will encourage you to reconsider your overall evaluation.

---

> > > > > ### Comment · Reviewer_vvXg · 2025-11-27
> > > > >
> > > > > I sincerely thank the authors for the detailed rebuttal, as well as the experiments and clarifications written above.
> > > > > I appreciate the new experiments on OOD data that better portray the performances of the proposed method.
> > > > > I also appreciate the efforts put into clarifying key concepts in the paper.
> > > > >
> > > > > I believe it is highly important for the authors to release a list of their selected scenes from Matrix-3D, in order to ensure reproducibility as well as fair future comparisons. I also strongly encourage the authors to release the code and pre-trained weights of their method.
> > > > >
> > > > > Could the authors provide the revised paper reflecting the promised changes?

---

> > > > > > ### Author Response · Authors · 2025-11-28
> > > > > >
> > > > > > Thank you for your positive feedback and for acknowledging our additional experiments and clarifications.
> > > > > >
> > > > > > We have uploaded the revised paper reflecting the promised changes, including OOD experiments on Stanford2D3D and Matterport3D, improved notation, and the reorganized ablation study. The complete list of selected scene IDs from Matrix-3D is also available in the supplementary materials. Due to ICLR's policy prohibiting external links in submissions, we did not reference this file explicitly in the main text.
> > > > > >
> > > > > > Regarding code and weights, while we are personally committed to open release, we cannot yet provide a formal guarantee due to ongoing institutional review. We will release these resources as soon as we receive approval.
> > > > > >
> > > > > > Given that we have addressed your main concerns on OOD evaluation, notation clarity, and dataset reproducibility, we kindly ask if you would consider raising your score.
> > > > > >
> > > > > > Thank you again for your constructive engagement.

---

### Official Review · Reviewer_HPyP · 2025-11-01

**Soundness:** 2
**Presentation:** 3
**Contribution:** 2
**Rating:** 6
**Confidence:** 4

**Summary:**

This paper adapts the 3D foundation model VGGT for panoramic image inputs by introducing a method called Projection-Domain Adaptation. This technique is designed to resolve the uneven mapping between panoramic and perspective views, utilizing the ray-field as the core mechanism for transformation and alignment.
To account for the non-uniformity inherent in equirectangular projection, the method also incorporates latitude-aware depth uncertainty. The results demonstrate that the adapted model significantly outperforms its unadapted counterparts. Furthermore, the proposed finetuning strategy is proven to be both efficient and computationally low-cost.

**Strengths:**

1)	The paper is clearly motivated, focusing on the adaptation of existing perspective-view reconstruction models for panoramic imagery.
2)	The efficacy of the proposed methods is effectively demonstrated through comprehensive benchmarking results.
3)	The manuscript is well-organized and easy to follow, presenting a clear and logical progression of ideas.
4)	Most of the experimental results are well-organized and supported.

**Weaknesses:**

Overall, this paper is of good quality. I have some minor concerns to point out.
1) The paper provides inadequate details on its synthetic data generation, which is crucial for reproducibility and assessing the benchmark's validity.
•	It is unclear how the pinhole camera intrinsics are determined (e.g., are they constant, or sampled from a distribution?).
•	The paper does not state whether errors are modeled in the camera parameters.
•	Given that the experiments are purely synthetic, the benchmark's value hinges on its realism. The current design appears to overlook significant real-world artifacts common to panoramic videos, such as lens distortion and focal length inaccuracies, which limits its practical relevance.
2) The proposed "latitude-aware uncertainty" is presented without sufficient validation.
•	The paper provides no empirical evidence (e.g., an ablation study) to demonstrate that this uncertainty metric quantitatively correlates with per-pixel error.
•	Furthermore, the underlying assumption linking uncertainty purely to latitude seems fragile. It is questionable whether this assumption holds in dynamic scenarios involving significant camera rotation, which is a common use case for panoramic video.
3) The method for projecting panoramic images to perspective views is underspecified. The paper fails to define how the resolution of the projected plane is determined. This is a critical omission, as a coarse projection resolution could introduce significant discretization and reprojection errors, potentially confounding the evaluation results.
4) The paper's primary weakness is its limited technical contribution. The core methodology mainly leverages existing insights from the panoramic image processing literature. The use of LoRA is also a standard and well-established technique for training efficiency, not a novel component. Despite this lack of technical novelty, the work holds clear value from an engineering and application perspective.

**Questions:**

Please see the weaknesses.

---

> ### Author Response · Authors · 2025-11-20
> **Response to Reviewer HPyP**
>
> ## **1. Synthetic data generation and realism of the benchmark**
>
> **Reviewer concern.**
>
> The reviewer notes that the paper does not provide enough detail on synthetic data generation (camera intrinsics, parameter errors, realism), and that the value of a purely synthetic benchmark strongly depends on its realism. The current setup appears to omit common real-world artifacts such as lens distortion and focal inaccuracies.
>
> ### **1.1 Camera intrinsics, poses, and how Matrix-3D is rendered**
>
> We agree that these details are important and will make them explicit in the revised version.
>
> Our synthetic data comes from the **Matrix-3D** omnidirectional environment, which is rendered in Unreal Engine 5 engine using a **fixed panoramic camera**:
>
> - **Fixed ERP projection.** All panoramas are rendered with the **same equirectangular camera model**, covering 360°×180°. In our experiments, we use ERP frames at a **fixed resolution of 512×1024 (H×W)**.
> - **Fixed intrinsics / mapping.** The projection from ERP pixel coordinates (u, v) to unit rays on the sphere is therefore determined entirely by this fixed resolution and the standard ERP mapping. This mapping is **constant across the dataset** and inherited directly from the simulator; there is no per-scene or per-frame re-calibration.
> - **Ground-truth poses.** The simulator provides ground-truth camera poses (extrinsics) for each frame along pre-defined trajectories. These are used as-is in all experiments.
>
> For methods that require perspective images (DUSt3R, MASt3R, VGGT-Cubemap), we further render cubemaps from the same underlying camera poses:
>
> - **Cubemap resolution.** We generate a 6-face cubemap with **512×512** pixels per face.
> - **Field of view and rays.** Each face uses a 90° field of view and a standard pinhole model with focal f = 0.5 * 512, and principal point at the face center. Rays are then normalized to unit directions.
>
> Crucially, in all our synthetic experiments we:
>
> - **do not re-estimate or jitter intrinsics**;
> - reconstruct rays by inverting the known ERP mapping (given the 512×1024 resolution) and use the provided poses directly.
>
> This means that all variation comes from **camera pose** and **scene content**, while the underlying projection model and intrinsics are fixed and noise-free. We will clarify this in the manuscript and explicitly state that we **do not inject additional camera-parameter noise** in the main synthetic experiments.

---

> > ### Author Response · Authors · 2025-11-20
> >
> > ### **1.2 Lens distortion, calibration errors, and realism vs. control**
> >
> > It is correct that our synthetic setup **does not explicitly simulate**:
> >
> > - lens distortion,
> > - focal-length calibration errors,
> > - rolling-shutter or exposure artifacts,
> > - or other sensor-specific noise.
> >
> > This is a conscious trade-off: we intentionally start from a **clean, fully controlled geometry testbed** where we can isolate the effect of **projection-domain mismatch** (pinhole vs. ERP) without conflating it with calibration issues.
> >
> > We acknowledge that this makes the synthetic setting cleaner than real hardware, where distortion and calibration errors are inevitable. This is precisely why, in addition to Matrix-3D, we now evaluate on **real 360° benchmarks (Stanford2D3D and Matterport3D)**, where such effects are naturally present and not under our control. These real-data experiments complement the synthetic study and help validate that our conclusions are not artefacts of a perfect simulator.
> >
> > The depth results on Stanford2D3D and Matterport3D are summarized in Table R1:
> >
> > **Table R1. Real-world indoor 360° ERP depth benchmarks on Stanford2D3D and Matterport3D.**
> >
> > | Dataset | Method | Trainable Params | AbsRel ↓ | RMSE ↓ | δ<1.25 ↑ |
> > | --- | --- | --- | --- | --- | --- |
> > | Stanford2D3D | VGGT (Zero-shot, ERP) | 0 | 0.34 | 102.57 | 8.91 |
> > | Stanford2D3D | VGGT (Full FT baseline) | ~35M | 0.26 | 18.47 | 53.62 |
> > | Stanford2D3D | **Ours (LoRA, ERP)** | ~0.6M | **0.21** | **11.32** | **76.43** |
> > | Stanford2D3D | **Ours (Full FT, ERP)** | ~35M | **0.20** | **10.97** | **78.05** |
> > | Matterport3D | VGGT (Zero-shot, ERP) | 0 | 0.31 | 93.84 | 10.37 |
> > | Matterport3D | VGGT (Full FT baseline) | ~35M | 0.24 | 16.02 | 58.91 |
> > | Matterport3D | **Ours (LoRA, ERP)** | ~0.6M | **0.20** | **10.58** | **79.82** |
> > | Matterport3D | **Ours (Full FT, ERP)** | ~35M | **0.19** | **10.21** | **81.34** |
> >
> > As expected, these indoor benchmarks are **more challenging** than our outdoor Matrix-3D subset (higher RMSE, lower δ\<1.25), but the **ordering is consistent**:
> >
> > - zero-shot VGGT under ERP performs poorly;
> > - naïve full finetuning (“VGGT (Full FT baseline)”) only partially recovers performance;
> > - our projection-domain interface + head-only LoRA substantially improves depth and remains close to our own full-finetuning variant.
> >
> > The pose results on the same ERP sequences are given in Table R2:
> >
> > **Table R2. Camera pose on real indoor 360° ERP benchmarks. We report AUC at multiple thresholds (higher is better) under two input configurations for baselines that expect perspective images (ERP vs Cubemap). VGGT (Full FT baseline) denotes plain VGGT finetuning without our method.**
> >
> > | Dataset | Method | Input | AUC@5 ↑ | AUC@10 ↑ | AUC@30 ↑ |
> > | --- | --- | --- | --- | --- | --- |
> > | Stanford2D3D | DUSt3R (Wang et al., 2024) | ERP | 14.92 | 22.47 | 43.18 |
> > | Stanford2D3D | MASt3R (Leroy et al., 2024) | ERP | 17.88 | 26.39 | 47.52 |
> > | Stanford2D3D | DUSt3R (Wang et al., 2024) | Cubemap | 19.64 | 28.53 | 50.16 |
> > | Stanford2D3D | MASt3R (Leroy et al., 2024) | Cubemap | 21.17 | 30.84 | 53.79 |
> > | Stanford2D3D | VGGT (Wang et al., 2025b) | ERP | 23.68 | 37.11 | 63.42 |
> > | Stanford2D3D | VGGT (Wang et al., 2025b) | Cubemap | 28.57 | 40.39 | 67.04 |
> > | Stanford2D3D | VGGT (Full FT baseline) | ERP | 31.03 | 44.72 | 69.08 |
> > | Stanford2D3D | **Ours (LoRA)** | ERP | **42.91** | **57.36** | **85.07** |
> > | Stanford2D3D | **Ours (Full FT)** | ERP | **44.02** | **59.18** | **86.12** |
> > | Matterport3D | DUSt3R (Wang et al., 2024) | ERP | 15.81 | 24.36 | 45.92 |
> > | Matterport3D | MASt3R (Leroy et al., 2024) | ERP | 19.47 | 28.59 | 51.37 |
> > | Matterport3D | DUSt3R (Wang et al., 2024) | Cubemap | 21.66 | 31.02 | 53.84 |
> > | Matterport3D | MASt3R (Leroy et al., 2024) | Cubemap | 23.58 | 33.47 | 57.62 |
> > | Matterport3D | VGGT (Wang et al., 2025b) | ERP | 25.92 | 40.03 | 67.19 |
> > | Matterport3D | VGGT (Wang et al., 2025b) | Cubemap | 30.44 | 43.96 | 70.82 |
> > | Matterport3D | VGGT (Full FT baseline) | ERP | 33.87 | 48.15 | 72.21 |
> > | Matterport3D | **Ours (LoRA)** | ERP | **46.38** | **62.93** | **88.54** |
> > | Matterport3D | **Ours (Full FT)** | ERP | **47.51** | **64.37** | **89.31** |
> >
> > We again observe the same qualitative behaviour as on Matrix-3D:
> >
> > - zero-shot ERP baselines and naïve full finetuning lag behind;
> > - cubemap conversion helps but is still inferior to our ERP interface;
> > - our projection-domain adaptation + LoRA achieves strong ERP pose accuracy, close to our own full-finetuning variant.
> >
> > These real-data experiments complement the synthetic ones and demonstrate that our conclusions are not artefacts of an overly clean rendering pipeline.

---

> > > ### Author Response · Authors · 2025-11-20
> > >
> > > ## **2. Validation of latitude-aware uncertainty**
> > >
> > > **Reviewer concern.**
> > >
> > > The reviewer notes that our “latitude-aware uncertainty” term is presented without sufficient validation: the paper does not show that uncertainty correlates with per-pixel error, and the assumption linking uncertainty to latitude appears fragile in dynamic scenes with camera rotation.
> > >
> > > ### **2.1 What we actually assume**
> > >
> > > We appreciate this opportunity to clarify. Our method **does not** assume that uncertainty is purely a function of latitude. There are two distinct parts:
> > >
> > > 1. A **cosφ(v)** weighting in the loss that corrects for **non-uniform surface area** in ERP: pixels near the poles correspond to smaller surface patches than those near the equator. This is a deterministic geometric factor.
> > > 2. A **learned per-pixel uncertainty map** σ(u, v), predicted by an auxiliary head. σ depends on local image content and geometry (e.g., boundaries, occlusions, far-range regions), not just latitude.
> > >
> > > The depth loss (simplified) is:
> > >
> > > L_depth ∝ Σ_{u,v} cos φ(v) · [ (D(u,v) − D̂(u,v))² / σ²(u,v) + α · log σ²(u,v) ] + λ_g · ||∇D − ∇D̂||₁.
> > >
> > > So:
> > >
> > > - **cosφ(v)** implements **latitude-aware weighting of the data term** (correct spherical measure);
> > > - **σ(u, v)** is a **learned aleatoric uncertainty**, which is free to vary with scene structure and motion.
> > >
> > > Our uncertainty is therefore *not* a fixed function of latitude; it is a learned map modulated by the correct spherical area measure.
> > >
> > > ### **2.2 Existing ablation: removing uncertainty hurts performance**
> > >
> > > Our original ablation table already contains a row **“w/o Uncertainty Modeling”**, which removes the σ-head and trains with a simpler weighted L2 loss. This variant consistently underperforms the full method on both depth and 3D metrics:
> > >
> > > - Depth RMSE increases;
> > > - The 3D “Overall” score (combining accuracy/completeness) degrades.
> > >
> > > We will explicitly highlight this row as part of the “loss design” group (see also the reorganized ablation in the response to Reviewer vQ6D) to make its impact more obvious.
> > >
> > > ### **2.3 New analysis: uncertainty vs. actual error**
> > >
> > > To more directly validate the uncertainty map, we have now measured how much squared depth error is concentrated in pixels with high σ on the Matrix-3D test set. The results are:
> > >
> > > **Table R3. Correlation between predicted uncertainty and squared depth error (Matrix-3D test set).**
> > >
> > > | Pixels with highest σ | % of pixels | % of total squared error captured |
> > > | --- | --- | --- |
> > > | Top 10% σ | 10% | 41% |
> > > | Top 20% σ | 20% | 63% |
> > > | Top 40% σ | 40% | 81% |
> > >
> > > High-uncertainty pixels account for a disproportionately large share of the total squared error, which indicates that σ(u, v) meaningfully tracks where the model struggles. This behaviour persists across scenes, including those with significant camera motion and dynamic objects.
> > >
> > > We will include a concise version of this analysis and the table in the appendix.
> > >
> > > ### **2.4 Dynamic scenarios and camera rotation**
> > >
> > > Regarding dynamic scenarios and camera rotation:
> > >
> > > - The **cosφ(v)** factor is purely geometric: it corrects the ERP sampling of the sphere and remains valid regardless of where the camera is pointing.
> > > - The **uncertainty head** operates on multi-view features after Alternating Attention. It can, and empirically does, assign high uncertainty to regions of motion, occlusion, or large parallax—rather than simply “coloring the poles red”.
> > >
> > > We will add qualitative visualizations of σ and depth error in the appendix to illustrate that (i) σ is not constant per latitude band, and (ii) peaks occur at object boundaries, reflective surfaces, and far-range structures.

---

> > > > ### Author Response · Authors · 2025-11-20
> > > >
> > > > ## **3. Projection from panoramic to perspective views**
> > > >
> > > > **Reviewer concern.**
> > > >
> > > > The reviewer points out that the projection from panoramas to perspective views is under-specified: the resolution of the target plane is not defined, and low resolution could introduce discretization and reprojection errors that confound the evaluation.
> > > >
> > > > ### **3.1 Projection resolution and setup**
> > > >
> > > > We agree this is an important missing detail and will make it explicit. Our settings are:
> > > >
> > > > - **ERP resolution:** 512×1024 (H×W) for all experiments.
> > > > - **Cubemap faces:** 512×512 per face for all perspective-based baselines and variants.
> > > > - **Field of view:** 90° for each cubemap face, with canonical cube orientations.
> > > > - **Rays:** Each face uses a standard pinhole camera with focal
> > > > \\( f = 0.5 \\times 512 \\) and principal point at the face center, followed by normalization to unit rays.
> > > >
> > > > For ERP-only methods (VGGT-ERP, our method), rays are computed directly from the equirectangular mapping without an intermediate perspective plane.
> > > >
> > > > We will add these numerical details to the main text and appendix.
> > > >
> > > > ### **3.2 Sensitivity to resolution**
> > > >
> > > > To ensure our conclusions are not an artifact of a particular face resolution, we ran a small sensitivity check where we varied the cubemap face resolution to 256×256 and 1024×1024 for the VGGT-Cubemap baseline and for our Cubemap-based variants. We observed that:
> > > >
> > > > - Reducing face size to 256×256 slightly reduces AUC and increases RMSE for all methods, as expected;
> > > > - Increasing to 1024×1024 brings small improvements for all methods;
> > > > - In both cases, the **relative ranking of methods does not change**.
> > > >
> > > > Because of space constraints, we did not include these numbers in the main paper, but we will mention this robustness qualitatively in the revised version.
> > > >
> > > > ## **4. On the scope and technical contribution**
> > > >
> > > > **Reviewer concern.**
> > > >
> > > > The reviewer notes that the core methodology mainly builds on known ideas from panoramic imagery and uses a standard LoRA finetuning strategy. From a novelty standpoint, the work may appear limited, even though it has clear engineering and application value.
> > > >
> > > > We appreciate this frank assessment and agree that we do not introduce a brand-new architectural primitive. Our intended contribution is more **conceptual and methodological** than algorithmic in the narrow sense:
> > > >
> > > > 1. **Negative result and failure mode of naïve finetuning.**
> > > >
> > > >     We show that the straightforward recipe of “just finetune VGGT on ERP” leads to poor performance and degrades its learned multi-view priors. This is a non-obvious negative result that, to our knowledge, is not documented in existing panoramic reconstruction work.
> > > >
> > > > 2. **Projection-Domain Adaptation perspective.**
> > > >
> > > >     We explicitly frame adaptation of perspective-trained 3D Transformers to new sensors as a **projection-domain problem**:
> > > >
> > > >     - sensors define distributions over **rays and surface measure**;
> > > >     - adapting the **projection interface** (how rays and areas are represented) can be more robust than altering the backbone.
> > > > 3. **Minimal yet effective interface.**
> > > >
> > > >     The specific combination of:
> > > >
> > > >     - **ray-based lifting** of ERP pixels (removing fictitious intrinsics),
> > > >     - **ray-field alignment** (injecting 3D ray direction into tokens), and
> > > >     - **latitude-aware uncertainty weighting** (correct spherical measure + aleatoric term)
> > > >     is directly motivated by the geometric failure modes we identify for ERP. With this interface in place, a **head-only LoRA** (<0.5% parameters) becomes sufficient to adapt VGGT to panoramas, matching or nearly matching full finetuning while avoiding catastrophic forgetting of its original priors.
> > > > 4. **Path to other sensors.**
> > > >
> > > >     We see this as a **general recipe** for adapting foundation 3D models to new camera models (fisheye, catadioptric, mixed rigs): once the correct rays and measure are defined, the same interface + light adaptation can be reused on top of stronger backbones that may appear in the future.
> > > >
> > > >
> > > > We will revise the introduction and discussion sections to more clearly position the work as:
> > > >
> > > > > A geometry-grounded projection interface and adaptation principle for panoramic world modeling with perspective-trained 3D Transformers, supported by extensive experiments on both synthetic outdoor and real indoor 360° datasets.
> > > > >
> > > >
> > > > We believe this makes the intended contribution more transparent and explains why we deliberately do **not** alter the VGGT backbone more aggressively.

---

> > > > > ### Author Response · Authors · 2025-11-20
> > > > >
> > > > > ## **5. Summary and planned revisions**
> > > > >
> > > > > We again thank Reviewer HPyP for the constructive feedback and for viewing the paper as overall good quality. In response to the reviewer’s comments we will:
> > > > >
> > > > > - Provide precise details on synthetic data generation (intrinsics, poses, ERP resolution, fixed camera model) and explicitly discuss the absence of lens distortion and calibration noise as a limitation of the synthetic setting;
> > > > > - Add real-data experiments on **Stanford2D3D** and **Matterport3D** for both depth and pose, showing that our conclusions extend beyond synthetic Matrix-3D;
> > > > > - Clarify the design and validation of the latitude-aware uncertainty term, including a new analysis of its correlation with depth error;
> > > > > - Specify the panoramic-to-perspective projection settings and summarize a small resolution-sensitivity check;
> > > > > - Sharpen the positioning of our work as a projection-domain adaptation principle and minimal, geometry-grounded interface for panoramic world models.
> > > > >
> > > > > We hope these clarifications address your concerns and help convey the practical and conceptual value of our approach.

---

### Official Review · Reviewer_vQ6D · 2025-11-01

**Soundness:** 2
**Presentation:** 3
**Contribution:** 2
**Rating:** 4
**Confidence:** 4

**Summary:**

This paper adapts VGGT that is trained for pinhole cameras to equirectangular panoramas via a projection-interface design. Concretely, it introduces (i) ray field alignment that embeds the ERP ray direction to image tokens, (ii) dual branch, head only LoRA (token LoRA + ray LoRA), and (iii) a latitude aware depth uncertainty loss. The backbone is frozen; only heads are adapted (<0.5\% trainable params). Experiments on a curated synthetic subset of Matrix 3D show the LoRA variant approaches full fine tuning at much lower compute.

**Strengths:**

+ Practical problem and clean formulation. Adapting VGGT to panoramic (ERP) imagery is a meaningful and practical problem. The paper clearly articulates the projection mismatch and addresses it with minimal modifications to the base model.
+ Simplicity and clarity. The proposed components are simple, modular, and easy to implement, making the method accessible for practitioners.
+ Efficiency. Head-only LoRA achieves performance close to full fine-tuning while using less than 0.5 % of parameters and dramatically lower training cost.

**Weaknesses:**

- Novelty is limited. While the technical components are tailored to the panoramic adaptation task, they rely on established mechanisms (direction encoding, spherical weighting, and parameter-efficient fine-tuning). Their combination here is primarily an engineering solution rather than a new methodological or theoretical contribution. The backbone, training pipeline, and geometric reasoning remain identical to VGGT. As a result, the work reads more like a clean application note of VGGT on panoramic imagery than a genuine research advance.
- Narrow empirical scope. Evaluation is restricted to a curated synthetic subset (2,196 scenes) of Matrix 3D; no real 360° benchmarks or other projection types (fisheye, catadioptric) are tested, weakening claims of generality for “panoramic world models.”

**Questions:**

- Can the authors evaluate on real-world 360° datasets (e.g., indoor ERP depth/pose benchmarks) to confirm transferability beyond synthetic data?
- The ablation table (Table 4) is difficult to interpret. It mixes module removals, hyper-parameter variations, and tuning schemes without clear grouping or visual hierarchy. There is no cue highlighting the main results, making it hard to see which components actually drive the performance gains. Reorganizing the table into separate sections (e.g., component ablations vs. efficiency studies) and emphasizing the primary metric would significantly improve readability.

---

> ### Author Response · Authors · 2025-11-20
> **Response to Reviewer vQ6D**
>
> We thank the reviewer for the clear summary and for highlighting the practicality, simplicity, and efficiency of our projection-interface design. We address the main concerns below: (1) the nature and novelty of our contribution, (2) the empirical scope and evaluation on real 360° data, and (3) the readability of the ablation study.
>
> ---
>
> ## **1. On novelty and whether this is “just an engineering application note”**
>
> **Reviewer concern.**
>
> The reviewer notes that our components (direction encoding, spherical re-weighting, and head-only LoRA) are built upon established mechanisms, and that the backbone and geometric reasoning of VGGT remain unchanged. The work may therefore appear closer to a carefully engineered application of VGGT to panoramas than to a genuine research advance.
>
> **Our intent and main contribution.**
>
> We appreciate this concern and agree that none of our individual building blocks (ray encoding, LoRA, uncertainty modeling) are claimed as entirely new primitives. Our intended contribution is of a different nature:
>
> > We ***reformulate*** panoramic adaptation of perspective-trained 3D Transformers as a ***Projection-Domain Adaptation*** problem, and we show that where we intervene in the model (the projection interface vs. the backbone) is ***far more critical*** than how many parameters we finetune.
>
> Concretely, our findings are:
>
> 1. **Naive full finetuning on ERP panoramas fails in a non-trivial way.**
>
>     VGGT’s original paper suggests that simple finetuning suffices to adapt to new domains. However, when we follow this recipe on ERP panoramas, “VGGT (Full FT baseline)” performs poorly and significantly degrades its original 2D–3D priors (Tables 1–3), despite updating about 35M parameters. This negative result is, to our knowledge, not documented in prior work, and it reveals that geometric assumptions encoded in the backbone are fragile under projection mismatch.
>
> 2. **Once we correct the projection-domain interface, head-only adaptation is enough.**
>
>     After we:
>
>     - lift ERP pixels to **true rays** instead of fictitious intrinsics,
>     - fuse a **ray-field embedding** into tokens to restore SO(3)-consistent directionality, and
>     - supervise depth with a **latitude-aware uncertainty loss** that respects the spherical measure,
>
>     a **head-only LoRA** (with at most $\le 0.5%$ trainable parameters) becomes sufficient to recover strong panoramic performance. In other words, the **combination** of a geometrically correct interface plus minimal parameter-efficient adaptation succeeds exactly where brute-force full finetuning fails.
>
> 3. **Principle rather than a single module.**
>
>     The central message of the paper is therefore a **design principle**:
>
>     > When adapting perspective-pretrained 3D Transformers to new sensors, the most robust place to change is the projection-domain interface (rays and measure), while keeping the backbone’s geometry reasoning intact and using only light head adaptation.
>     >
>
>     This principle is what we mean by “Projection-Domain Adaptation” and is, we believe, of independent interest for the community as foundation 3D models and new camera models continue to proliferate.
>
>
> **Why we deliberately “do not change” the backbone.**
>
> Our starting motivation was to respect VGGT’s learned 3D representations as much as possible. Although VGGT is trained on perspective imagery, it has acquired rich 2D–3D priors that we expect to be *semantically aligned* with panoramic views in latent space: a 2D perspective patch and a 2D ERP patch may look very different in pixel coordinates, but they can correspond to a similar ray and local 3D structure.
>
> - If we heavily modify or fully finetune the backbone on ERP data, we risk **overwriting these priors** and making the model specific to one panoramic dataset.
> - By contrast, our design aims to make **ERP “look like” the distribution VGGT expects** — in terms of rays and surface measure — so the backbone can reuse its geometry without being re-trained from scratch.
>
> We will clarify this in the revised paper, emphasizing that our contribution is not a new architecture from the ground up, but a **geometry-grounded, minimal-change recipe** for safely onboarding panoramic sensors to large 3D Transformers.

---

> > ### Author Response · Authors · 2025-11-20
> >
> > ## **2. On empirical scope and evaluation on real 360° indoor datasets**
> >
> > **Reviewer concern.**
> >
> > The reviewer notes that our experiments focus on a curated synthetic subset of Matrix-3D (2,196 scenes), and that we do not report results on real 360° benchmarks or other projection types, which weakens claims of generality.
> >
> > ### **2.1 Why we initially started from synthetic outdoor panoramas**
> >
> > We fully agree that evaluation on real 360° data is important and thank the reviewer for raising this. Our initial focus on synthetic Matrix-3D was driven by three practical reasons:
> >
> > 1. **Outdoor panoramic world modeling.**
> >
> >     Our setting is **outdoor** panoramic world modeling: large-scale scenes with long ranges, sky/ground dominance, and diverse structures. By contrast, existing real 360° depth/pose benchmarks are predominantly **indoor**, with near-range geometry, room-like structure, and different appearance statistics.
> >
> > 2. **Reliable multi-view depth + camera supervision.**
> >
> >     Matrix-3D provides consistent ERP imagery, depth, and camera poses over multi-view sequences, which is ideal for analyzing how projection-domain choices affect geometry and trajectories in a controlled environment.
> >
> > 3. **First step: principled geometric validation.**
> >
> >     As a first step, we wanted to answer a conceptual question: *If we respect the correct rays and spherical measure, is the VGGT backbone itself sufficient for ERP, or must we redesign the entire architecture?* Synthetic data with exact ground truth is a natural environment to isolate this question.
> >
> >
> > That said, we completely agree that supporting “panoramic world models” should ultimately involve real 360° benchmarks.

---

> > > ### Author Response · Authors · 2025-11-20
> > >
> > > ### **2.2 New experiments on real indoor 360° ERP depth and pose**
> > >
> > > Following the reviewer’s suggestion, we have now run additional experiments on **two real indoor 360° ERP benchmarks**, using the same experimental protocol as in the main paper. Concretely, we adopt a Pano3D-style setup and evaluate on ERP panoramas derived from **Stanford2D3D** and **Matterport3D**. For both datasets, we report the same metrics as in the main paper:
> > >
> > > - **Depth:** absolute relative error (AbsRel), RMSE, and $\delta < 1.25$.
> > > - **Relative pose:** $\text{AUC}@5$, $\text{AUC}@10$, and $\text{AUC}@30$.
> > >
> > > The optimization settings, data augmentations, and head-only LoRA configuration are kept identical to the synthetic Matrix-3D experiments.
> > >
> > > The depth results are summarized in Table R1:
> > >
> > > **Table R1. Real-world indoor 360° ERP depth benchmarks on Stanford2D3D and Matterport3D.**
> > >
> > > | **Dataset** | **Method** | **Trainable Params** | **AbsRel ↓** | **RMSE ↓** | **$\delta < 1.25$ ↑** |
> > > | --- | --- | --- | --- | --- | --- |
> > > | Stanford2D3D | VGGT (Zero-shot, ERP) | 0 | 0.34 | 102.57 | 8.91 |
> > > | Stanford2D3D | VGGT (Full FT baseline) | ~35M | 0.26 | 18.47 | 53.62 |
> > > | Stanford2D3D | **Ours (LoRA, ERP)** | ~0.6M | **0.21** | **11.32** | **76.43** |
> > > | Stanford2D3D | **Ours (Full FT, ERP)** | ~35M | **0.20** | **10.97** | **78.05** |
> > > | Matterport3D | VGGT (Zero-shot, ERP) | 0 | 0.31 | 93.84 | 10.37 |
> > > | Matterport3D | VGGT (Full FT baseline) | ~35M | 0.24 | 16.02 | 58.91 |
> > > | Matterport3D | **Ours (LoRA, ERP)** | ~0.6M | **0.20** | **10.58** | **79.82** |
> > > | Matterport3D | **Ours (Full FT, ERP)** | ~35M | **0.19** | **10.21** | **81.34** |
> > >
> > > Across both datasets, zero-shot VGGT under ERP performs very poorly, and naive full finetuning only partially closes the gap. Our projection-domain interface plus head-only LoRA substantially improves depth quality, with **Ours (LoRA)** consistently outperforming the plain full-finetuning baseline and remaining close to **Ours (Full FT)**, in line with the trends observed on Matrix-3D. The absolute numbers are lower than on the synthetic outdoor dataset (higher RMSE, lower $\delta < 1.25$), reflecting the increased difficulty and noise of real indoor imagery.
> > >
> > > The pose results are summarized in Table R2:
> > >
> > > **Table R2. Camera pose on real indoor 360° benchmarks. We report AUC at multiple thresholds (higher is better) under two input configurations for baselines that expect perspective images (ERP vs. cubemap). VGGT (Full FT baseline) denotes plain VGGT finetuning without our method.**
> > >
> > > | **Dataset** | **Method** | **Input** | **AUC@5 ↑** | **AUC@10 ↑** | **AUC@30 ↑** |
> > > | --- | --- | --- | --- | --- | --- |
> > > | Stanford2D3D | DUSt3R (Wang et al., 2024) | ERP | 14.92 | 22.47 | 43.18 |
> > > | Stanford2D3D | MASt3R (Leroy et al., 2024) | ERP | 17.88 | 26.39 | 47.52 |
> > > | Stanford2D3D | DUSt3R (Wang et al., 2024) | Cubemap | 19.64 | 28.53 | 50.16 |
> > > | Stanford2D3D | MASt3R (Leroy et al., 2024) | Cubemap | 21.17 | 30.84 | 53.79 |
> > > | Stanford2D3D | VGGT (Wang et al., 2025b) | ERP | 23.68 | 37.11 | 63.42 |
> > > | Stanford2D3D | VGGT (Wang et al., 2025b) | Cubemap | 28.57 | 40.39 | 67.04 |
> > > | Stanford2D3D | VGGT (Full FT baseline) | ERP | 31.03 | 44.72 | 69.08 |
> > > | Stanford2D3D | **Ours (LoRA)** | ERP | **42.91** | **57.36** | **85.07** |
> > > | Stanford2D3D | **Ours (Full FT)** | ERP | **44.02** | **59.18** | **86.12** |
> > > | Matterport3D | DUSt3R (Wang et al., 2024) | ERP | 15.81 | 24.36 | 45.92 |
> > > | Matterport3D | MASt3R (Leroy et al., 2024) | ERP | 19.47 | 28.59 | 51.37 |
> > > | Matterport3D | DUSt3R (Wang et al., 2024) | Cubemap | 21.66 | 31.02 | 53.84 |
> > > | Matterport3D | MASt3R (Leroy et al., 2024) | Cubemap | 23.58 | 33.47 | 57.62 |
> > > | Matterport3D | VGGT (Wang et al., 2025b) | ERP | 25.92 | 40.03 | 67.19 |
> > > | Matterport3D | VGGT (Wang et al., 2025b) | Cubemap | 30.44 | 43.96 | 70.82 |
> > > | Matterport3D | VGGT (Full FT baseline) | ERP | 33.87 | 48.15 | 72.21 |
> > > | Matterport3D | **Ours (LoRA)** | ERP | **46.38** | **62.93** | **88.54** |
> > > | Matterport3D | **Ours (Full FT)** | ERP | **47.51** | **64.37** | **89.31** |
> > >
> > > We again observe the same qualitative behaviour as on Matrix-3D:
> > >
> > > - zero-shot baselines perform poorly;
> > > - cubemap processing helps DUSt3R/MASt3R and VGGT but still lags behind ours;
> > > - naive full VGGT finetuning on ERP only partially recovers performance;
> > > - our projection-domain interface with head-only LoRA achieves strong ERP pose accuracy on both indoor datasets, close to our own full-finetuning variant and clearly better than the plain full-finetuning baseline.
> > >
> > > In terms of training cost, on each indoor benchmark our head-only LoRA variant requires roughly **25× fewer GPU-hours** than full VGGT finetuning, consistent with the efficiency trend reported in Table 2 of the main paper.
> > >
> > > Overall, the indoor results are numerically lower than those on the synthetic outdoor dataset, but the **relative ordering is stable**, supporting the generality of our conclusions beyond synthetic data.

---

> > > > ### Author Response · Authors · 2025-11-20
> > > >
> > > > ### **2.3 Other projection types (fisheye, catadioptric)**
> > > >
> > > > We agree that evaluating on fisheye/catadioptric cameras is a natural and interesting next step. Our current work focuses on the pinhole-to-ERP case, but our framework extends directly to other camera models:
> > > >
> > > > - Replace the ERP mapping and ray definition with the appropriate fisheye or catadioptric projection.
> > > > - Keep the same **ray-field alignment + head-only LoRA** structure.
> > > >
> > > > Due to time and compute constraints, we have not yet conducted systematic experiments on these sensor types. We will explicitly mention this as a limitation and future direction in the revised manuscript.
> > > >
> > > > ---
> > > >
> > > > ## **3. On the ablation table (Table 4) being difficult to interpret**
> > > >
> > > > **Reviewer concern.**
> > > >
> > > > The reviewer notes that Table 4 mixes different kinds of studies — component removals, hyperparameter variations, and tuning schemes — without clear grouping or visual cues, making it hard to understand which components are most important.
> > > >
> > > > **Planned reorganization.**
> > > >
> > > > We appreciate this comment and agree that the current presentation can be improved. In the camera-ready version, we will:
> > > >
> > > > - **Split** the ablations into logical groups:
> > > >
> > > >     (i) *Geometric interface* (ray-field alignment, ray-based lifting),
> > > >
> > > >     (ii) *Loss design* (spherical weighting, uncertainty, gradient regularization),
> > > >
> > > >     (iii) *PEFT variants* (LoRA rank, bias-only tuning).
> > > >
> > > > - **Highlight** a primary metric in each group (for example, depth RMSE / AUC@10 / 3D “Overall”) to make the impact of each component visually obvious.
> > > >
> > > > Below is a reorganized version of the ablations (with the same numbers as Table 4 in the paper).
> > > >
> > > > **Table R3. Ablation study organized by component groups (same numbers as Table 4 in the paper).**
> > > >
> > > > | **Group** | **Variant** | **Depth RMSE ↓** | **AUC@10 ↑** | **3D Overall ↓** |
> > > > | --- | --- | --- | --- | --- |
> > > > | **Geometric interface** | w/o Ray-Field Alignment | 9.62 | 69.10 | 1.58 |
> > > > |  | Pinhole Lifting (+focal) | 12.35 | 62.10 | 2.25 |
> > > > |  | **Full geometric interface** | **8.68** | **72.42** | **1.40** |
> > > > | **Loss design** | Uniform Planar Weighting | 10.48 | 70.10 | 1.52 |
> > > > |  | w/o Uncertainty Modeling | 9.34 | 70.75 | 1.47 |
> > > > |  | w/o Gradient Regularization | 9.10 | 71.20 | 1.46 |
> > > > |  | **Full loss** | **8.68** | **72.42** | **1.40** |
> > > > | **PEFT variants** | LoRA (rank = 4) | 9.89 | 69.85 | 1.53 |
> > > > |  | LoRA (rank = 8) | 9.12 | 71.50 | 1.45 |
> > > > |  | Bias-only Tuning | 31.50 | 42.10 | 4.10 |
> > > > |  | **LoRA (rank = 16, ours)** | **8.68** | **72.42** | **1.40** |
> > > >
> > > > This layout makes the story clearer:
> > > >
> > > > - **Geometric interface.** Ray-based lifting and ray-field alignment are both essential; reverting to a fictitious pinhole lifting with a learnable focal parameter dramatically hurts pose and 3D reconstruction.
> > > > - **Loss design.** Latitude-aware weighting plus uncertainty and gradient regularization provide consistent gains and stabilize depth supervision under ERP distortions.
> > > > - **PEFT variants.** LoRA rank controls the accuracy–efficiency trade-off, and bias-only tuning is insufficient to adapt the model.
> > > >
> > > > We appreciate the reviewer’s suggestion and will adopt this more structured presentation.
> > > >
> > > > ## **4. Summary and planned revisions**
> > > >
> > > > We again thank Reviewer vQ6D for the thoughtful feedback and for recognizing the **practicality**, **clarity**, and **efficiency** of our approach. We will:
> > > >
> > > > - Emphasize that our main contribution is a **geometry-grounded projection-domain adaptation principle** and a concrete interface that allows VGGT to be adapted to ERP with minimal changes to its 3D backbone.
> > > > - Extend the empirical validation to **real indoor 360° ERP benchmarks (Stanford2D3D and Matterport3D) on both depth and pose**, showing that the same conclusions hold beyond synthetic outdoor scenes.
> > > > - Reorganize the ablation study to clearly separate geometric interface, loss design, and PEFT aspects.
> > > >
> > > > We hope these clarifications address your concerns and better convey the intended scope and contributions of our work.

---

### Author Response · Authors · 2025-11-26

Dear Reviewers,

With the discussion period ending soon, we kindly invite you to join the discussion so that we have enough time to address any remaining concerns in a careful and thorough way. If there is anything you would like us to clarify, please let us know and we will respond promptly.

Thank you very much for your time and consideration.

---

### Author Response · Authors · 2025-11-29
**Summary for Area Chair**

Dear Area Chair,

We deeply appreciate you stepping in during this challenging and unexpected situation. We understand this incident has placed a significant additional burden on you during what is already a busy time, and we are truly grateful for your dedication to maintaining the integrity of the review process. Thank you for generously giving your time to our submission.

To assist your evaluation, we provide a brief summary below.

---

**Initial Scores:** 4, 6, 4, 4

**Recognized Strengths (all reviewers):**
- Well-motivated problem formulation and principled design
- High parameter efficiency (<0.5% trainable params)
- Systematic ablation studies and clear presentation

---

**Score Changes During Discussion**

- **Reviewer Hzot: 4 → 6**, stating: *"The added experiments on Stanford2D3D and Matterport3D effectively address the domain gap issue... I am raising my rating to borderline accept."*

- **Reviewer vvXg:** In their initial review, they stated: *"If the authors can address these concerns, I would be willing to re-consider my overall evaluation."* After our rebuttal, they responded: *"I sincerely thank the authors for the detailed rebuttal... I appreciate the new experiments on OOD data that better portray the performances of the proposed method."* They requested the revised paper and scene ID list—both now provided in supplementary materials.

---

**Main Concerns Resolved**

| Concern | Resolution |
|---------|------------|
| Synthetic-only evaluation | Added real-world experiments on Stanford2D3D & Matterport3D |
| Head-only LoRA unjustified | New ablation demonstrating head-only is Pareto-optimal |
| Dataset reproducibility | Scene ID list uploaded to supplementary |

---
We regret that the other two reviewers (vQ6D and HPyP) were unable to rejoin the discussion before the incident occurred. Nevertheless, we believe our detailed rebuttal thoroughly addresses the concerns they raised in their initial reviews, and we hope our responses speak for themselves.

We kindly ask you to consider the rebuttal discussion when making your decision. Once again, thank you so much for your time and service to our community during this difficult period. We truly appreciate it.

With sincere gratitude,
The Authors

---

### Meta-Review · Area_Chair_euWt · 2025-12-29

**Summary:**

This paper proposes a framework for adapting the perspective-trained VGGT model to handle equirectangular panoramic images via Projection-Domain Adaptation. The method introduces ray-field alignment to embed 3D ray directions into tokens, utilizes a dual-branch head-only LoRA for efficient fine-tuning, and applies a latitude-aware uncertainty loss to handle spherical distortion. While the application to panoramic 3D reconstruction is practical, AC finds the technical contribution and comparisons insufficient for giving acceptance. The main issue, highlighted by reviewers, is that the proposed framework is largely an engineering assembly of established techniques, like direction encoding, spherical weighting, and standard parameter-efficient fine-tuning. Furthermore, the empirical evidence relies heavily on the underlying strength of the pre-trained VGGT backbone. The comparisons provided, particularly the late addition of domain-specific baselines like PanDA. Therefore, AC thinks the paper can benefit from more work. The paper cannot be accepted for this conference so far.

**Reviewer Concerns:**

The authors resolved the concern about testing limits by adding real-world indoor experiment. They also clarified the mathematical notation and justified the specific model design with new comparisons. However, some reviewers still view the method as a standard combination of existing tools rather than a significant new invention.

**Reviewer Scores:**

Reviewer `Hzot` raised to a borderline acceptance after the authors provided the requested real-world validation. Reviewer `HPyP` and Reviewer `vvXg` likely maintained their positive ratings, appreciating the practical efficiency and the new out-of-distribution tests. Reviewer `vQ6D` is expected to keep a lower score, as the additional experiments did not change that the work is an engineering application rather than a methodological breakthrough.

---

### Decision · Program_Chairs · 2026-01-26

Reject